# Long-lived central memory γδ T cells confer protection against murine cytomegalovirus reinfection

**Nathalie Yared[1], Maria Papadopoulou[2,3,4], Pierre Barennes[5], Hang-Phuong Pham[5], Valentin Quiniou[5], Sonia Netzer[1], Hanna Kaminski[1], Laure Burguet[1], Amandine Demeste[1], Pacôme Colas[1], Lea Mora-Charrot[6], Benoit Rousseau[6], Julien Izotte[6], Atika Zouine[7], Xavier Gauthereau[8], David Vermijlen[2,3,4,9], Julie Déchanet-Merville[1‡], Myriam Capone[1‡]***

1 Bordeaux University, Centre National de la Recherche Scientifique, Institut National de la Santé et de la Recherche Médicale, ImmunoConcEpt, UMR 5164, ERL 1303, ImmunoConcEpt, Bordeaux, France, 2 Department of Pharmacotherapy and Pharmaceutics, Université Libre de Bruxelles (ULB), Brussels, Belgium, 3 Institute for Medical Immunology, Université Libre de Bruxelles (ULB), Gosselies, Belgium, 4 Université Libre de Bruxelles Center for Research in Immunology, Université Libre de Bruxelles (ULB), Brussels, Belgium, 5 Parean Biotechnologies, Saint-Malo, France, 6 Bordeaux University, Service Commun des Animaleries, Bordeaux, France, 7 Bordeaux University, Centre National de la Recherche Scientifique, Institut national de la santé et de la recherche médicale, FACSility, TBM Core, Bordeaux, France, 8 Bordeaux University, Centre National de la Recherche Scientifique, Institut national de la santé et de la recherche médicale, OneCell, RT-PCR and Single Cell Libraries, TBM Core, Bordeaux, France, 9 WELBIO department, Walloon ExceLlence Research Institute, Wavre, Belgium

‡ These authors are co-senior authors on this work.
* mcapone@immuconcept.org

**Data Availability Statement:** The authors confirm that all data underlying the findings are fully available without restriction. All relevant data are

## Abstract

The involvement of γδ TCR-bearing lymphocytes in immunological memory has gained increasing interest due to their functional duality between adaptive and innate immunity. γδ T effector memory (TEM) and central memory (TCM) subsets have been identified, but their respective roles in memory responses are poorly understood. In the present study, we used subsequent mouse cytomegalovirus (MCMV) infections of αβ T cell deficient mice in order to analyze the memory potential of γδ T cells. As for CMV-specific αβ T cells, MCMV induced the accumulation of cytolytic, KLRG1+CX3CR1+ γδ TEM that principally localized in infected organ vasculature. Typifying T cell memory, γδ T cell expansion in organs and blood was higher after secondary viral challenge than after primary infection. Viral control upon MCMV reinfection was prevented when masking γδ T-cell receptor, and was associated with a preferential amplification of private and unfocused TCR δ chain repertoire composed of a combination of clonotypes expanded post-primary infection and, more unexpectedly, of novel expanded clonotypes. Finally, long-term-primed γδ TCM cells, but not γδ TEM cells, protected T cell-deficient hosts against MCMV-induced death upon adoptive transfer, probably through their ability to survive and to generate TEM in the recipient host. This better survival potential of TCM cells was confirmed by a detailed scRNASeq analysis of the two γδ T cell memory subsets which also revealed their similarity to classically adaptive αβ CD8 T cells. Overall, our study uncovered memory properties of long-lived TCM γδ T cells that confer protection in a chronic infection, highlighting the interest of this T cell subset in vaccination approaches.

within the paper and its Supporting Information files.

**Funding:** This work was supported by the Agence Nationale de la Recherche (https://anr.fr/received, grant number ANR-19-CE18-0024-02 to JDM) and by the Fondation pour la Recherche Médicale (https://www.frm.org/, to JDM). The funders had no role in study design, data collection and analysis, decision to publish, or preparation of the manuscript. The funders had no role in study design, data collection and analysis, decision to publish, or preparation of the manuscript".

**Competing interests:** The authors have declared that no competing interests exist.

## Author summary

Cytomegalovirus (CMV) is a widespread, latent virus that can cause severe organ disease in immune-compromised patients. Anti-CMV memory immune responses are essential to control viral reactivation and/or reinfection events that commonly take place in solid organ transplantation. The role of γδ T-cell receptor bearing lymphocytes could be crucial in this context where immunosuppressive/ablative treatments cause suboptimal and/or delayed αβ T cell responses. Here we asked whether γδ T cells could compensate for the absence of αβ T cells in the long-term control of mouse CMV infection. Three months post-primary viral challenge in αβ-T cell deficient mice, γδ T cells displayed similar features as cytolytic, CMV-specific αβ CD8 T cells. We showed that previous priming with CMV endowed γδ T cells with an enhanced antiviral potential and that long-term maintenance of γδ T cell-mediated antiviral protection was dependent on γδ central memory T cells (TCM). As observed in human, the γδ T cell response to a secondary CMV challenge generated a private TCR δ repertoire. Finally, our gene expression/accessibility single cell analysis revealed that γδ TCM shared similar features as their αβ T cell counterpart. Our results sustain the adaptive-like properties of these unconventional T cells and reveal the interest of targeting γδ TCM subset in novel antiviral vaccination approaches.

## Introduction

The concept of immunological memory has been challenged in recent years. It was described as an acquired, multidimensional and evolutionary process that can no longer be solely related to vertebrates and adaptive immunity. With regards to the host capacity to improve survival upon secondary infection, immunological memory implicates both the innate and adaptive arm of the immune system (reviewed in [1–3]).

Prototypical adaptive memory cells are CD8+ αβ T lymphocytes, composed of various populations harboring distinct migratory, proliferative and effector properties: CCR7+CD62L+CD27+ Central Memory T cells (TCM), mostly found in secondary lymphoid organs, and CCR7-CD62L-CD27- Effector Memory T cells (TEM), among which CX3CR1+ circulating TEM and CX3CR1-CD103+CD49a+CD69+ Tissue-Resident Memory T cells (TRM). In the case of resolving infections, CD8 TEM progressively decline unless submitted to antigenic re-challenge. When the pathogen persists, these cells are retained and acquire specific features reminiscent to the nature of infection. In chronic infections (LCMV clone 13 in mice or HIV, HCV in humans), antiviral CD8 TEM display an increased expression of inhibitory receptors such as PD-1. In latent infections however (herpesviruses), such "exhausted" phenotype is uncommonly found, at least among circulating CD8 TEM (For reviews see [4–7]).

Among latent β-herpesviruses, cytomegaloviruses (CMVs) have drawn great attention due to the deleterious effects of CMV infection in immune-compromised individuals. In healthy subjects, remodeling of the CD8+ αβ TCR repertoire occurs over time during latency, with an increased frequency of cells specific for a few viral epitopes, a phenomenon referred to as memory inflation. Long-term CMV-induced CD8 T cells express a TEM KLRG1+ phenotype. In humans, they use the longer CD45 isoform (CD45RA), reminiscent of highly differentiated cells. Inflationary CD8+ αβ T cells circulate in the blood, home to peripheral tissues and show increased expression of cytolytic proteins. They are assumed to maintain robust function and prevent viral spread upon reactivation (for review see [8–11]). The crucial role of long-lasting memory to human CMV (HCMV) is exemplified in solid organ transplantation (SOT), where

seronegative recipients (R-) receiving a CMV+ graft (D+) are at higher risk of developing CMV disease than seropositive recipients (R+).

Another population expressing somatically diversified, T-cell antigen (Ag) receptors (TCR) is delineated by MHC-unrestricted, γδ TCR-bearing lymphocytes, which recognize ubiquitous stress-induced self ligands [12]. Their implication in immunological memory has been put forward in recent years, in different settings including pathogen infection and autoimmunity (reviewed [13–15]). Because of their dual nature, γδ TCR-mediated memory responses display both innate-like and adaptive-like characteristics [16–18]. In fact, γδ T cells are composed of so-called innate-like subsets with invariant and public (shared between individuals) TCR repertoires, as well as adaptive-like subsets with unfocused and highly private TCR repertoires. Innate-like γδ T lymphocytes are preprogrammed before birth and thus display rapid effector function in periphery. On the other hand, adaptive-like γδ T cells (which comprise mouse Vγ1+ and Vγ4+ and human non-Vγ9Vδ2 T cell subsets) exit the thymus after birth with a naïve phenotype. Reviewed in [19–21].

Some years ago, our group identified HCMV as a major driver of non-Vγ9Vδ2 clonal expansions in peripheral blood of renal transplant recipients, that was further on confirmed by other teams in different settings including allo-hematopoietic stem cell transplantation (HSCT) [22] (reviewed in [23,24]). A γδ T cell contribution to the anti-viral response was suggested by the concomitant diminution of viremia [25,26]. The protective role of γδ T cells was evidenced thanks to the mouse model. Concomitantly to the group of Winkler, we showed that γδ T cells can compensate for the absence of αβ T cells and confer protection to mouse CMV (MCMV) (in TCRα$^{-/-}$ or CD4-depleted CD8$^{-/-}$JHT mice) [27,28]. In TCRα$^{-/-}$ mice, the acute phase response to MCMV engaged Vγ1+ and Vγ4+ T cells that acquired a CD44+ CD62L- (TEM) phenotype alongside infection resolution. To which extent and for how long previous contact with CMV endows γδ T cells with an increased antiviral reactivity remains to be clarified.

As observed for HCMV-specific inflationary CD8+ αβ T lymphocytes, long-term HCMV-induced non-Vγ9Vδ2 T cells express a TEMRA (CD45RA+CD27-CD62L-CCR7-) KLRG1 + phenotype. Moreover, transplant patients experiencing a secondary infection (D-R+, *i.e.* HCMV-seropositive recipients receiving graft from a HCMV-seronegative donor) show a faster recall response of TEMRA γδ T cells and better infection resolution comparatively to patients experiencing a primary infection (D+R-) [26,29]. These results strongly suggest that γδ T cells have a memory potential against CMV. In this study, we addressed this issue using our previously set up TCRα$^{-/-}$ mouse model that allows experimental re-infection and adoptive transfer experiments. Our results show the preferential amplification of private TRD (TCR δ chain) repertoire upon MCMV reinfection, and highlight the crucial role of CD44+CD62L+ γδ TCM for long-term maintenance of γδ-mediated antiviral protection.

## Results

### 1. MCMV leaves a long-lasting imprint on γδ T cells

To test the long-term impact of MCMV on γδ T cell phenotype and function, we used αβ T cell deficient mice, in which we previously demonstrated the protective antiviral role of γδ T cells [27]. Mice were kept infected for three months, and γδ T cells analyzed in the blood as well as in liver and lung, that are main sites of MCMV infection and where γδ T cells expand [27]. Based on previous knowledge on αβ T cells, we used CD44 and CD62L expression to characterize γδ TCM-like (CD44+CD62L+) and TEM-like (CD44+CD62L-) subsets, that will be further on referred to as γδ TCM and TEM for simplification. Long-term MCMV infection induced an increase of γδ CD44+CD62L- TEM percentages concomitant to a decrease of

CD44+CD62L+ TCM, in lung, liver and blood (Fig 1A and Tab A in S1 Data) while, as we observed earlier [27], few naïve (CD44-CD62L+) γδ T cells were evidenced even in control uninfected mice.

We then assessed the presence of KLRG1 and CX3CR1 on γδ TEM from day 92 (d92) infected mice, because of their reported expression on inflationary CMV-specific CD8+ αβ T cells [30–32]. As shown in S1A Fig, KLRG1 and CX3CR1 were co-expressed on γδ TEM, while KLRG1 and PD1 showed mutually exclusive expression (S1B Fig). In samples from MCMV-naïve mice, 20–40% of γδ TEM were KLRG1+CX3CR1+ in blood and lung (Fig 1B, upper panels), while γδ TCM were KLRG1-CX3CR1- in all sites (Fig 1B, lower panels). However, whereas γδ TCM remained mostly KLRG1-CX3CR1- in d92-infected mice, the proportion of KLRG1+CX3CR1+ cells among γδ TEM inflated upon infection (Fig 1B and Tab A in S1 Data).

To extend these data, we compared immune-related genes expressed by γδ T cells from organs of uninfected *vs* MCMV-infected mice (S2 Fig and Tab A in S2 Data). Among the transcripts that were increased concomitantly in the liver and lungs at d92 (S2A Fig), were found genes encoding proteins associated with cytotoxicity (Granzyme A and B, Perforin), inhibitory receptors (NKG2A and NKG2C), as well as CX3CR1, M-CSF and CMKLR1 (chemerin chemokine-like receptor 1). In contrast, we evidenced a decrease of *Ccr7* and *Cdkn1a* (cyclin dependent kinase inhibitor 1) transcripts (S2A Fig). Cell surface analysis of granzyme A (grA) further showed that KLRG1+CX3CR1+ γδ TEM from target organs of long-term infected mice exhibit higher cytolytic potential, comparatively to age-matched control mice (Fig 1C and Tab A in S1 Data).

Taken together, these results demonstrate that, as reported for αβ T cells, MCMV leaves a long-lasting imprint on γδ T cell phenotype and function, characterized by a raise of cytotoxic KLRG1+CX3CR1+ TEM cells in infected organs and blood.

## 2. γδ TEM co-expressing KLRG1 and CX3CR1 preferentially localize in organ vasculature

To gain a better knowledge on the localization of γδ T cell memory effectors likely involved in the local/tissue antiviral control, we discriminated circulating *versus* tissue resident γδ T cell memory subsets using a well-validated labeling technique involving intravenous (i.v.) infusion of fluorescently conjugated anti-CD45 antibody (Ab). This allows staining of circulating T cells whereas T cells retained within tissues are protected from Ab binding [33]. The majority of γδ T cells from the lung and liver of d92-infected mice appeared to be CD45+, thus circulating (IV+) (Fig 2A). This repartition was not a consequence of infection since similar results were obtained in control mice. Interestingly, γδ TCM from infected lung were mostly found in the IV+ fraction (Fig 2B). CX3CR1+KLRG1+ γδ TEM were also largely enriched in the IV + fraction (Fig 2C and 2D), in accord with preferential expression of CX3CR1 on peripheral memory T cells (TPM) [34]. Pulmonary "resident" (IV-) γδ TEM cells were indeed mainly composed of CX3CR1-KLRG1- cells expressing PD1 (Fig 2D and Tab B in S1 Data). When comparing infected and uninfected mice, our results reveal a tendency toward higher CX3CR1 +KLRG1+ and lower CX3CR1-KLRG1-PD1+ γδ TEM in both IV+ and IV- cells from liver and lung (Fig 2D).

Thus, long-term local/tissue γδ T cell response to MCMV infection probably involves both circulating and resident cells, but the most prevalent subset are circulating γδ TEM co-expressing KLRG1 and CX3CR1.

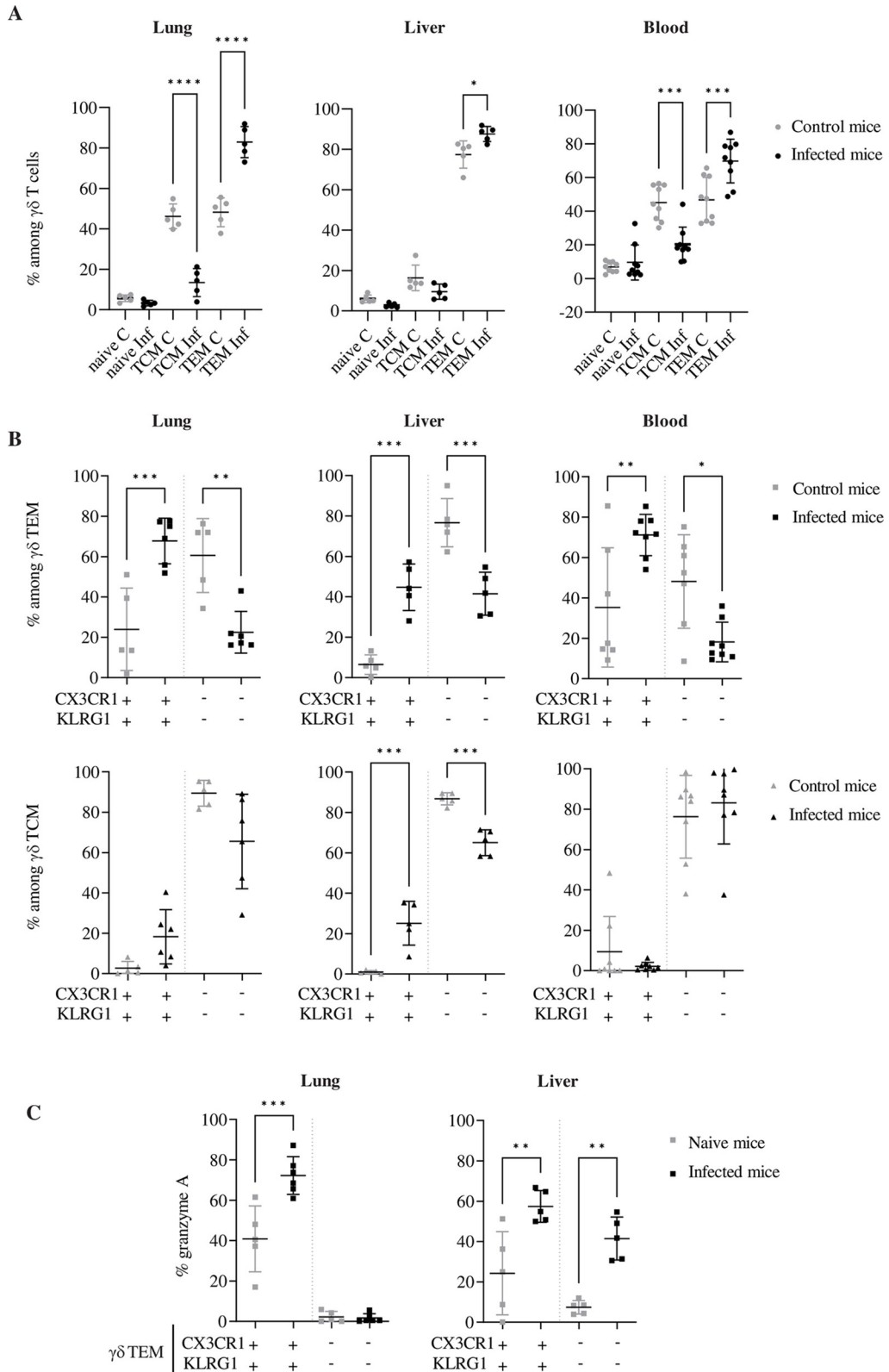

**Fig 1. MCMV leaves a long-lasting imprint on γδ T cell phenotype and function.** Long-term (d92) TCRα-/- MCMV infected mice (dark) were compared to uninfected TCRα-/- mice (grey). (A) Percentages of naïve (CD44-CD62L+), TCM (CD44+CD62L+) and TEM (CD44+CD62L-) cells among γδ T lymphocytes from indicated samples in individual mice. (B) Percentages of KLRG1+CX3CR1+ and KLRG1-CX3CR1- cells among γδ TEM (upper panels) and among γδ

TCM (lower panels). (C) Intracellular Granzyme A expression in gated CX3CR1+KLRG1+ and CX3CR1-KLRG1- γδ TEM populations from lung and liver. Data represent mean+/- SD and are representative of 2 independent experiments. Statistical test was 1-way ANOVA.

## 3. Reduction of γδ TCM slightly affects viral control in organs from long-term MCMV-infected mice

Fingolimod (FTY720) has been regularly used as a blocker of T cell egress from lymph node. In the setting of MCMV infection, this drug was used to show that maintenance of MCMV-specific effector CD8 T cells does not depend on migration through, or antigen recognition within the lymph [35]. When applying short-term (11 days) FTY720 treatment to long-term MCMV-infected mice, a pronounced decrease of γδ TCM counts was evidenced in blood, lung and liver comparatively to untreated TCRα$^{-/-}$ mice (Fig 3A, upper panels and Tab C in S1 Data). In contrast, γδ TEM numbers remained mostly constant in organs despite treatment, although they were reduced in blood (Fig 3A, lower panel and Tab C in S1 Data). Thus, as for αβ T cells, FTY720 primarily affects re-circulation of γδ TCM. After treatment, viral loads slightly increased in organs from d92-infected mice, although significance was only reached for liver (p = 0,022) (Fig 3B and Tab C in S1 Data). These results suggest that γδ TCM may somehow participate to the control of viral loads, although the latter principally relies on γδ TEM in the short term.

## 4. γδ T cell response to secondary MCMV challenge is faster and higher than during primary infection

All above results demonstrate a long-term imprint of MCMV on γδ T cells highly suggestive of their memory potential. To assess whether γδ T cells develop a memory response against MCMV, we used subsequent infections of αβ T cell deficient mice. In order to converge to analyses done in humans, we initially monitored the kinetics of γδ T cell memory subtypes longitudinally, in blood samples taken post primary and secondary MCMV-challenge (see experimental scheme, Top S3A Fig). In accordance with previous findings [28,36], the first phase of primary infection was marked by a drop in numbers of all γδ T cell differentiation subtypes, likely related to virus-driven lymphopenia (Fig 4A). Then, γδ T cell counts increased to reach a peak at day 21 where they exceeded basic levels (d0) for TEM. After second viral challenge, maximal counts of γδ TEM were obtained at day 7, i.e. much earlier than after first viral encounter (Fig 4A and Tab D in S1 Data; S3A Fig, bottom left and Tab B in S2 Data).

A second set of experiments was performed to analyze γδ T cell memory response to MCMV in organs: TCRα$^{-/-}$ mice were sacrificed at different time points following secondary MCMV challenge, while the γδ T cell response to primary infection was analyzed simultaneously in age-matched mice (see experimental scheme, S3B Fig). Soon after secondary viral challenge, γδ TEM counts doubled in organs. When compared to primary infection, a more rapid and higher increase of γδ TEM was achieved, with statistical differences observed at day 1 and 7 for lungs and day 1 for liver (Fig 4B and Tab D in S1 Data). By the mean time, cytotoxicity-related gene expression in γδ T cells quickly increased after reinfection, especially in the lungs (S2B Fig and Tab A in S2 Data). Finally, given the high proportion of KLRG1+ cells among MCMV-induced γδ TEM (Fig 1B), KLRG1+ γδ TEM dominated the γδ T cell secondary response to MCMV in blood (S4A Fig and Tab C in S2 Data; S3A Fig, bottom right and Tab B in S2 Data) and organs (S4B Fig and Tab C in S2 Data).

In their whole, our data depict a more rapid and higher γδ T cell response to MCMV after secondary challenge comparatively to the primary response, which are hallmarks of T cell memory.

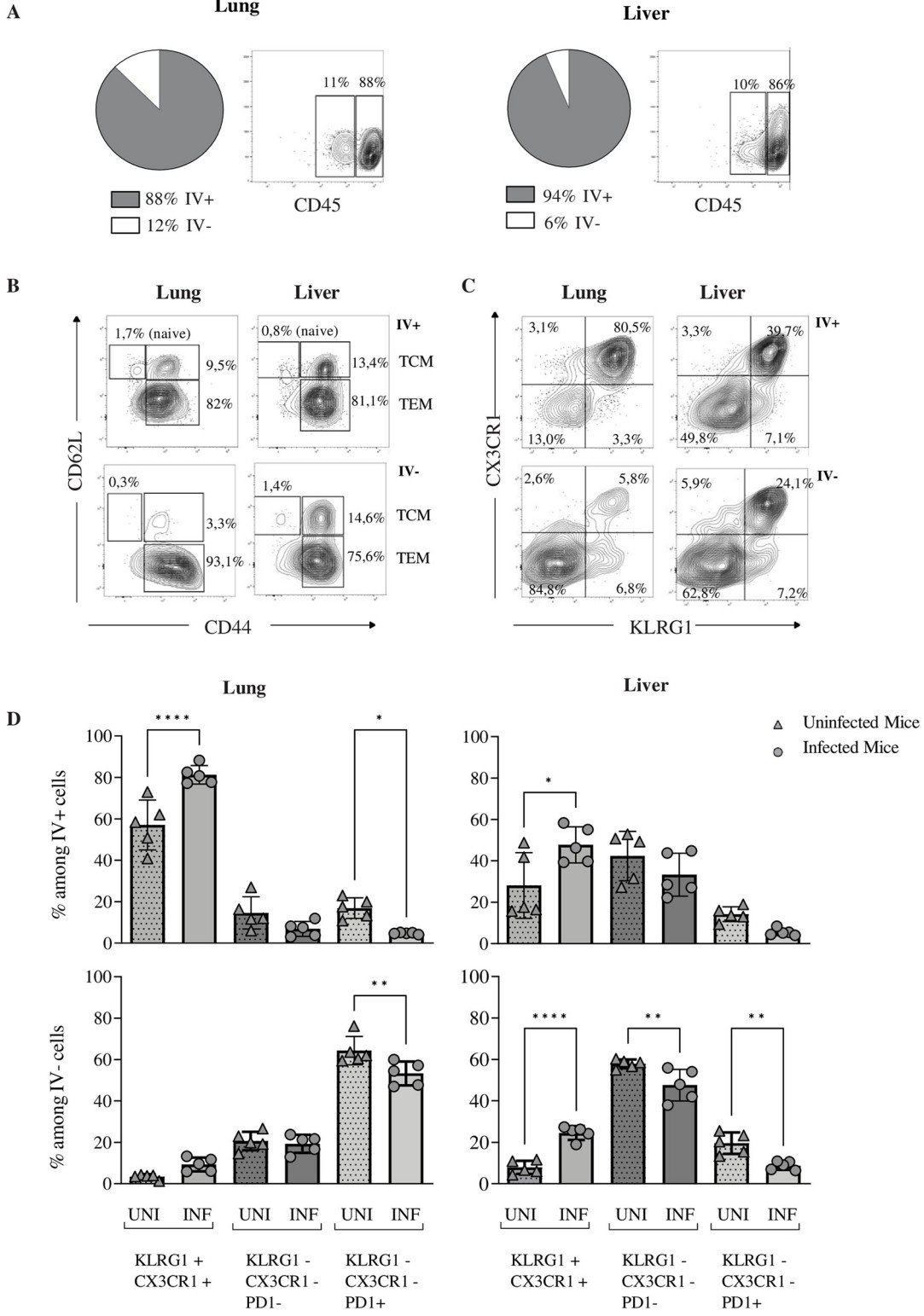

**Fig 2. γδ T cells expressing KLRG1/CX3CR1 preferentially localize in the vasculature.** Long-term MCMV infected TCRα-/- mice (INF) (n = 5) or naïve age-matched control mice (UNI) (n = 5) received anti-CD45 labeled mAb intravenously. Mice were sacrificed 5 min after antibody injection. (A) Repartition of γδ T cells in the intravascular positive (IV+) or intravascular negative (IV-) fraction in indicated organs from infected mice. Pie charts show the mean percentages of γδ T cells from 5 mice in each fraction. Representative contour plots obtained for one mouse out of 5 are shown. (B) Repartition of γδ

naïve, TCM and TEM in the IV+ or the IV- fraction from organs of one representative infected mouse. (C) Repartition of γδ TEM subtypes within the IV+ or IV- compartment according to KLRG1 and CX3CR1 expression. (D) Comparative analyses between control uninfected (triangle) or infected (circle) mice, of the repartition of the γδ T cell memory subtypes within the IV+ or IV- compartment, according to the expression of KLRG1, CX3CR1 and PD1. Data represent the mean +/- SD of proportions of γδ T cell memory subsets from 5 infected mice or 5 naïve mice in one representative experiment out of 2. Statistical test was 1-way ANOVA.

## 5. Viral control after reinfection is affected by anti-γδ TCR treatment

To confirm the participation of γδ T cells in the secondary antiviral response, long-term infected TCRα$^{-/-}$ mice received intravenous injection of the anti-γδ TCR GL3 mAb or isotypic control, followed (24h later) by MCMV secondary challenge (Fig 5A). GL3 treatment was proposed to induce down regulation of the cell surface expression of the TCR, rendering γδ T cells invisible and poor responders to TCR-specific stimulation [37]. The effect of the anti-γδ TCR mAb treatment was evidenced by the absence of cell surface expression of TCRδ (Fig 5B, left). γδ T cells treated *in vivo* with GL3 were nevertheless still producing IFNγ upon *in vitro* activation by PMA and ionomycin indicating their conserved ability to respond to TCR-independent activation (Fig 5B, right and Tab E in S1 Data), although other functions (such as cytotoxicity) might be affected. Seven days post-re-infection, a substantial increase of viral loads was observed in organs from mice that had received the anti-γδ TCR mAb comparatively to control mice (Fig 5C and Tab E in S1 Data). In marked contrast, during primary infection, the effect of GL3 mAb was obtained only when mice were treated during the third week of infection but not when GL3 treatment was given during the first or the second week of infection (Fig 5D and Tab E in S1 Data).

Collectively, our data indicate that at least two weeks of priming are required for γδ T cell antiviral function to be active. They put forward the accelerated impact of GL3-treatment on viral load control in the course of secondary *versus* primary infection, once again illustrating γδ T cell memory.

## 6. MCMV drives the expansion of a private and adaptive-like TCRδ repertoire

As the γδ TCR is involved in the viral control by long-term-induced γδ T cells in our model, we tested whether there were changes in the TRD (TCRδ chain) repertoire upon MCMV infection. We performed TRD CDR3 NGS analysis on blood at different time points post-primary (short-term (d21), long-term (d80)) and post-secondary (d7R) MCMV infection. At d7R, the TRD repertoires of lung, liver and spleen were analyzed in parallel. Age-matched control mice were studied simultaneously to take into account possible impact of ageing [38]. No major differences were observed between uninfected (n = 3) and infected mice (n = 5) in TRDV usage, CDR3 length distribution and number of N additions (S5A Fig and Tab D in S2 Data).

While the TRD repertoire appeared to be relatively stable between d21 and d80, obvious changes were observed after infection (d21) and reinfection (d7R) (S6A Fig). Some clonotypes present at d0 (dark blue, mice 1 and 2) were barely found at d21, leaving space to other clonotypes (see for example mouse 1, where clone CALMEREAFWGGELSATDKLVF (TRDV2-2) became highly prominent). The secondary γδ T cell response to MCMV was composed by a combination of clonotypes expanded post-primary infection and of novel expanded clonotypes (for example mouse 2 and 3, S6A Fig in orange). Note that at d7R a higher TRD blood repertoire overlap with organs was observed compared to d0 (S7 Fig and Tab E in S2 Data), indicating that the blood γδ T cell response to MCMV reflects (at least partially) what occurs in organs at a given time point.

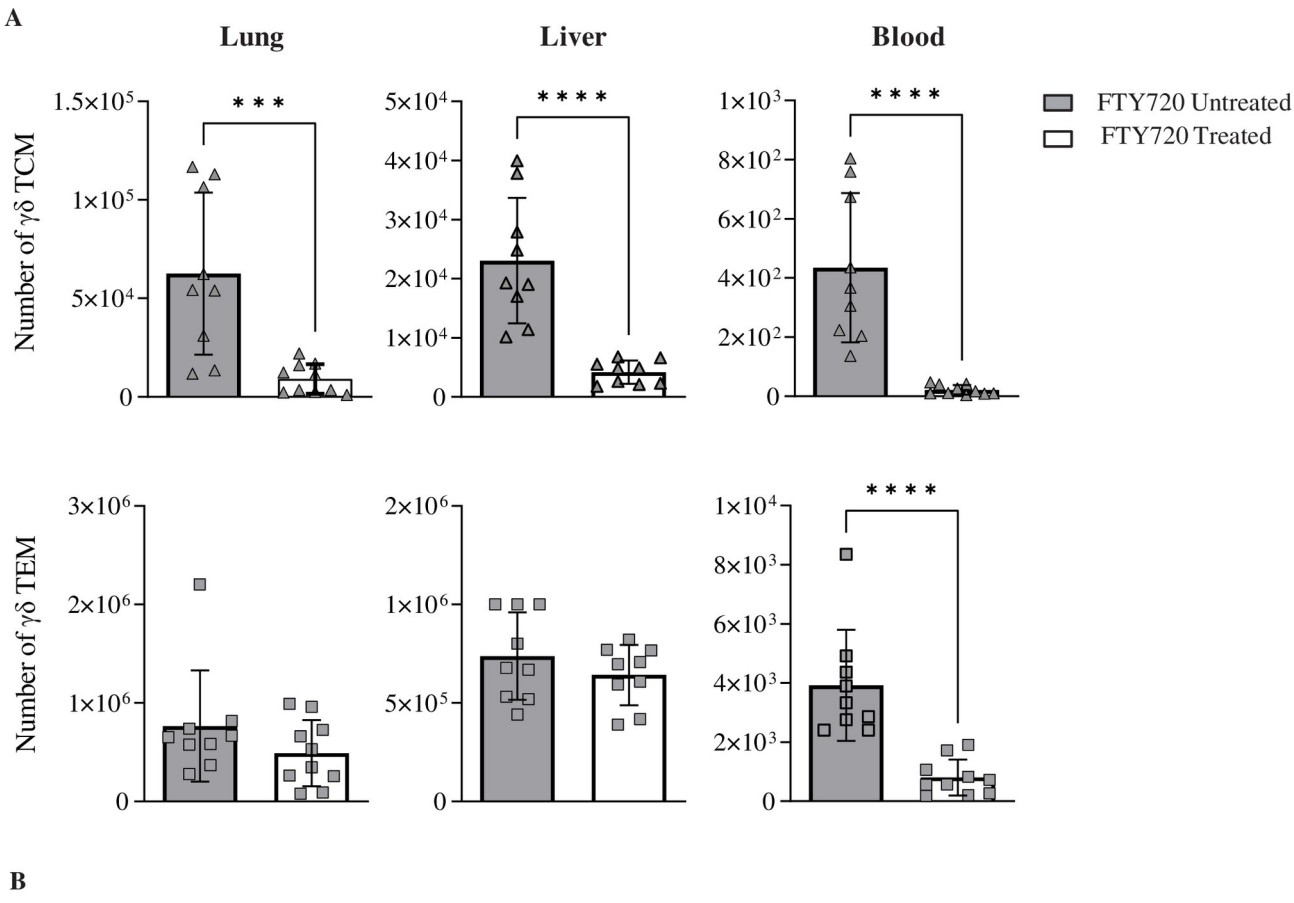

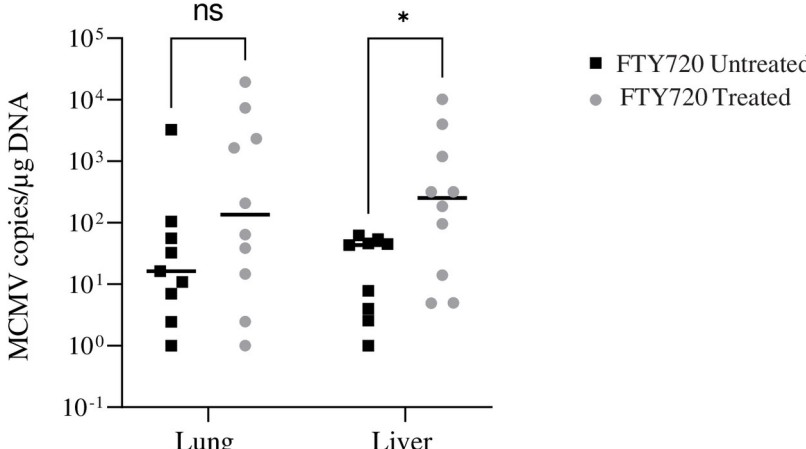

**Fig 3. Reduction of γδ TCM slightly affects viral load control in organs from long-term infected mice.** Long-term MCMV infected TCRα-/- mice (n = 10) were treated (n = 5) or not (n = 5) with FTY720 for 11 days before sacrifice. (A) Histograms show the number of γδ TCM (upper panels) and TEM (lower panels) in organs (total numbers) or blood (per 25 μl). (B) Viral loads in organs of treated or untreated mice were quantified by qPCR. Each point indicates one individual mouse and represents MCMV copy numbers in 1 μg of total DNA. Data are pooled from two independent experiments (n = 5 per group). (A-B) Error bars represent the standard error of the mean of γδ T cell numbers (A) and of the median of MCMV copy numbers (B). Differences were evaluated by the Mann-Whitney statistical test.

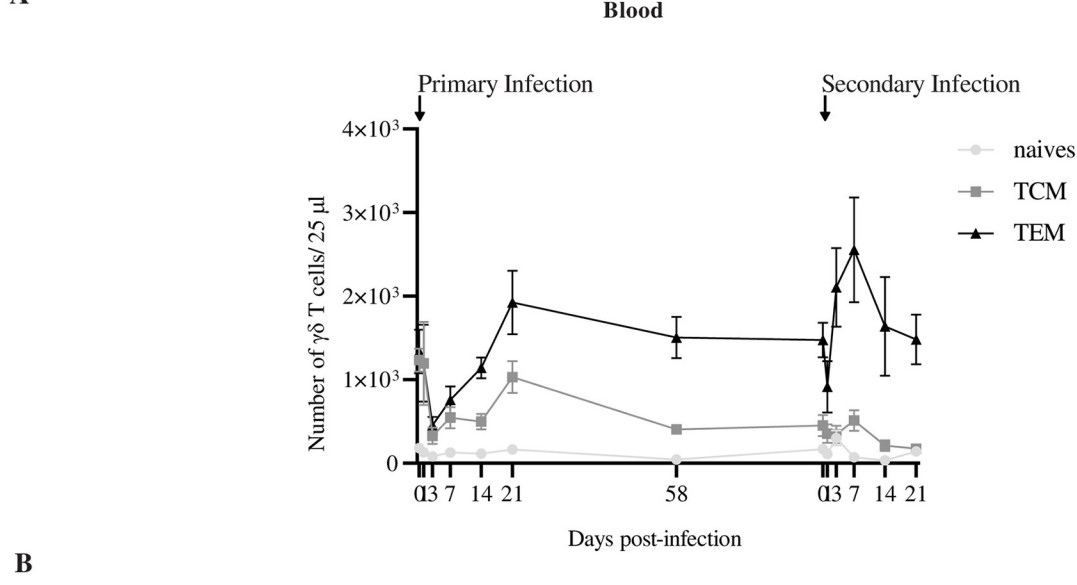

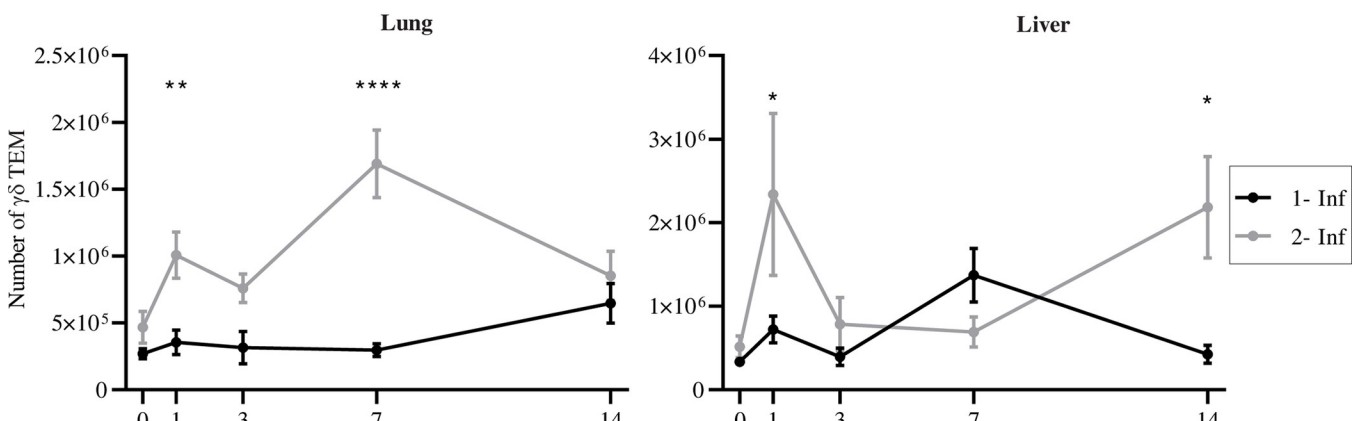

**Fig 4. γδ T cell response to secondary MCMV challenge.** (A) Longitudinal analysis of γδ T cells in blood following the experimental scheme depicted in S3A Fig. Mean absolute numbers +/- SEM of γδ T lymphocytes in 25 μl of blood are shown. (B) Analysis in organs following the experimental scheme depicted in S3B Fig. Shown are total number of γδ T lymphocytes in organs. Data represent the mean +/- SEM of cell counts from 5 mice sacrificed at indicated days post-primary (1- Inf, black) and secondary (2- Inf, grey) infection. (A, B) 2-way ANOVA test was used for comparison. These experiments were repeated twice with comparable results.

In comparison to infected mice, the kinetic evolution of clonotypes in blood of uninfected mice remained more stable overall, although some changes also occurred consecutively to control medium injection (S6B Fig). Notably however, the frequency of shared clonotypes between individuals of the infected mice group (i.e. F value, a repertoire overlap measure) was lower than that of the control group (Fig 6A and Tab F in S1 Data), while no clear influence could be observed on the diversity of the repertoire (S5B Fig and Tab D in S2 Data), indicating that the TRD repertoire became more private upon MCMV infection (unique to each individual). Among the public sequences highly shared between samples, we identified the innate CDR3 amino acid sequences CGSDIGGSSWDTRQMFF (*TRDV4* sequence) [39–42] and CALWEPHIGGIRATDKLVF (TRAV15-1-DV6-1 sequence) related to so-called NK T γδ [43,44] (for nomenclature see [45]). In line with an increased private repertoire upon infection, a tendency towards lower representation of these shared clonotypes in infected *versus* uninfected mice was observed (Fig 6B and Tab F in S1 Data).

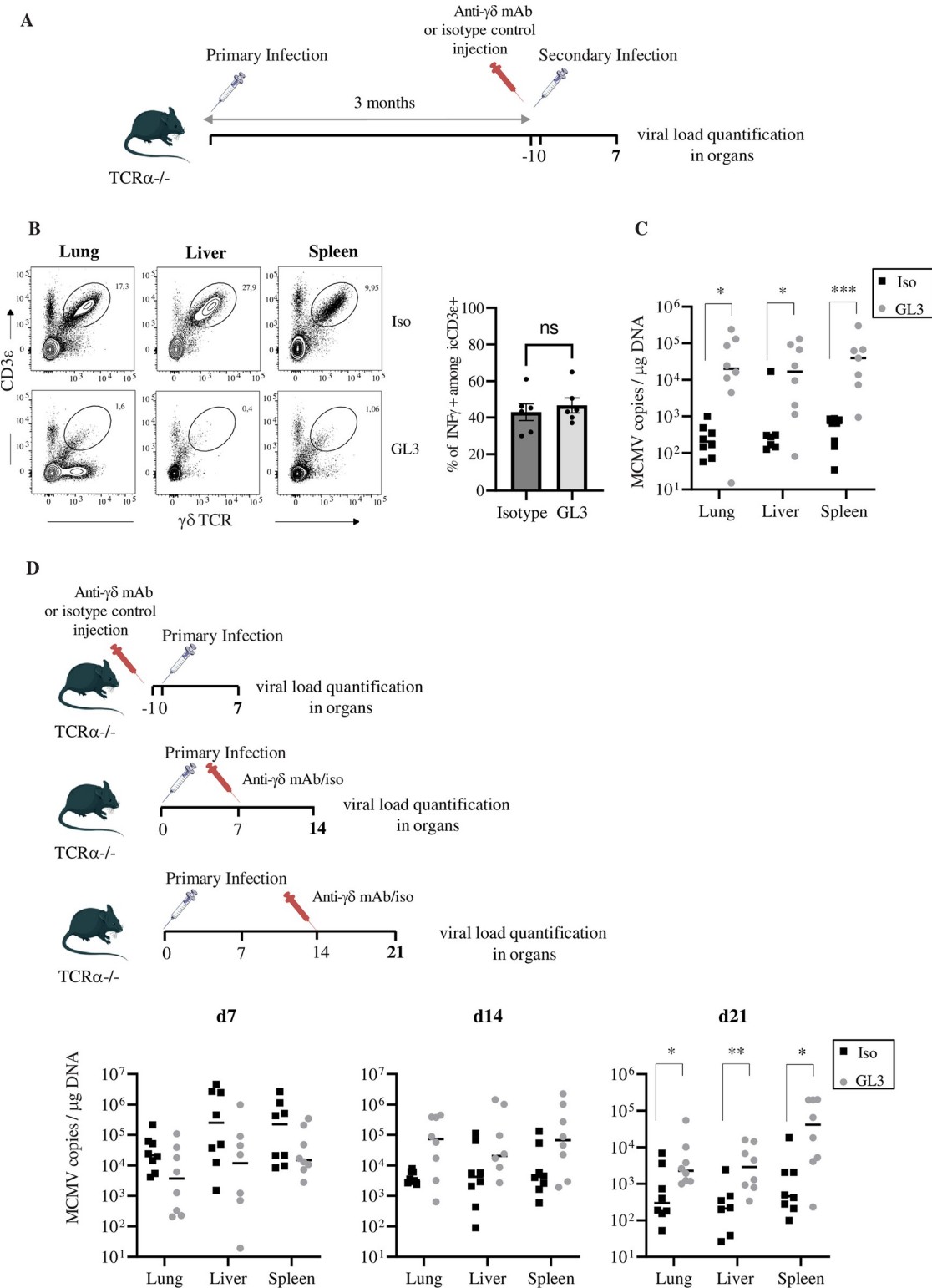

**Fig 5. Viral control is affected by anti-γδ TCR treatment.** (A) Experimental scheme. TCRα-/- mice (n = 10) were infected with MCMV (2.10³ PFU) at day 0. At day 92, mice received isotype control or anti-γδ mAb (n = 4). 24h after, mice were re-challenged with similar doses of MCMV. (B, left) Staining of CD3ε+/ γδ+ cells in mice treated with anti-γδ TCR (GL3) or control mAb (iso) at day 7 post reinfection. (B, right) Percentage of IFNγ+ among intracellular (ic) CD3ε+ splenocytes from 6 GL3-treated and 6 control mice, after 3h incubation with PMA-ionomycin. (C) Viral loads/1 µg of total DNA for individual mouse were quantified in organs

by qPCR 7 days post-reinfection. The experiment was repeated twice with comparable results. Results were pooled in the figure. Statistical tests were Mann Whitney. (D) TCRα-/- mice (n = 24) were infected with MCMV ($2.10^3$ PFU) at day 0. At day -1, 7 or 14, mice received isotype control (iso) or anti-γδ mAb (GL3), and were sacrificed at day 7, 14 or 21, respectively (n = 4/condition). Viral loads/1 μg of total DNA for individual mouse were quantified in organs by qPCR. The experiment was repeated twice with comparable results. Results were pooled in the figure. Statistical test was Mann Whitney. Images of mice and syringes were drawn by using pictures from Servier Medical Art. Servier Medical Art by Servier is licensed under a Creative Commons Attribution 3.0 Unported License (https://creativecommons.org/licenses/by/3.0/).

In αβ T cells, it has been demonstrated that CDR3aa (Antigen Binding specific Region), differing by only one amino acid, can recognize the same peptide [46–48]. This CDR3aa sequence proximity allows for the clustering of CDR3aa sequences that confer the same antigenic specificity to T cells. Employing this method, we tested whether γδ TCR repertoire at d7R (pooling blood, liver, lung and spleen) clustered more with d0 or with d21 γδ TCR repertoire. In infected mice, we observed a significant increase in mean cluster size for d21/d7R compared with d0/d7R (Fig 6C and S1 Table). This increase was not observed in uninfected mice. This closer amino acid sequences of CDR3aa between d21 and d7R versus between d0 and d7R only in infected mice, suggested a shared γδ TCR response against CMV antigens during primary and secondary infection.

In sum, our results highlight the preferential amplification and/or recruitment of a private and adaptive-like TRD repertoire upon MCMV infection. The γδ T cell secondary response to MCMV appears to involve both γδ TCR clonotypes expanded after the primary MCMV encounter as well as novel expanded clonotypes.

## 7. Long-term MCMV-induced γδ T cells confer protection to T cell deficient hosts upon adoptive transfer

As shown earlier, adoptive transfer of short-term (2–3 weeks) MCMV-primed γδ T cells confer protection to T cell deficient hosts [27,28]. However, the repartition and phenotype of γδ T cell memory subtypes importantly changed along the course of MCMV infection, with an increase of KLRG1 and other inhibitory receptors (Figs 1 and S2B). To test whether long-term MCMV-primed γδ T cells conserve their protective antiviral function, we isolated splenic γδ T cells from three month-MCMV-infected TCRα$^{-/-}$ mice, and transferred them into CD3ε$^{-/-}$ hosts (Fig 7A). γδ T cells isolated from MCMV-infected mice significantly increased the survival rate of CD3ε$^{-/-}$ hosts to MCMV infection performed one day after the transfer (Fig 7B, left). Improved viral control in γδ-T cell bearing *vs* T-cell deficient mice was evidenced by lower viral loads in organs (Fig 7B, right and Tab G in S1 Data). To assess the importance of MCMV priming on γδ T cells prior to their transfer in recipient mice, γδ-T cells were also sorted from age-matched control uninfected mice (5 months old) and transferred into T-cell deficient hosts. By contrast to MCMV-primed γδ T cells, "naïve" γδ T cells (i.e. unprimed by MCMV) did not protect T cell deficient hosts from death (Fig 7C, upper left) and did not control viral loads at sacrifice (Fig 7C, upper right and Tab G in S1 Data). Yet, splenic γδ T cells from long-term infected and age-matched TCRα$^{-/-}$ mice comprise similar proportions of memory subtypes (S8 Fig and Tab F in S2 Data). Enhanced MCMV disease symptoms in naïve versus MCMV-primed γδ -T cell CD3ε$^{-/-}$ bearing mice was attested by higher quantities of transaminases (Fig 7C, lower left panel and Tab G in S1 Data). Alongside, mice transferred with unprimed γδ T lymphocytes had lower γδ T cell counts (Fig 7C, lower right panel and Tab G in S1 Data). Finally, we analyzed γδ T cells in CD3ε$^{-/-}$ mice that had survived for 130 days after transfer with MCMV-primed γδ T cells. The repartition of γδ T cell memory subtypes in γδ-bearing CD3ε$^{-/-}$ and d92 infected TCRα$^{-/-}$ mice was comparable, evidencing efficient reconstitution (S9 Fig and Tab G in S2 Data).

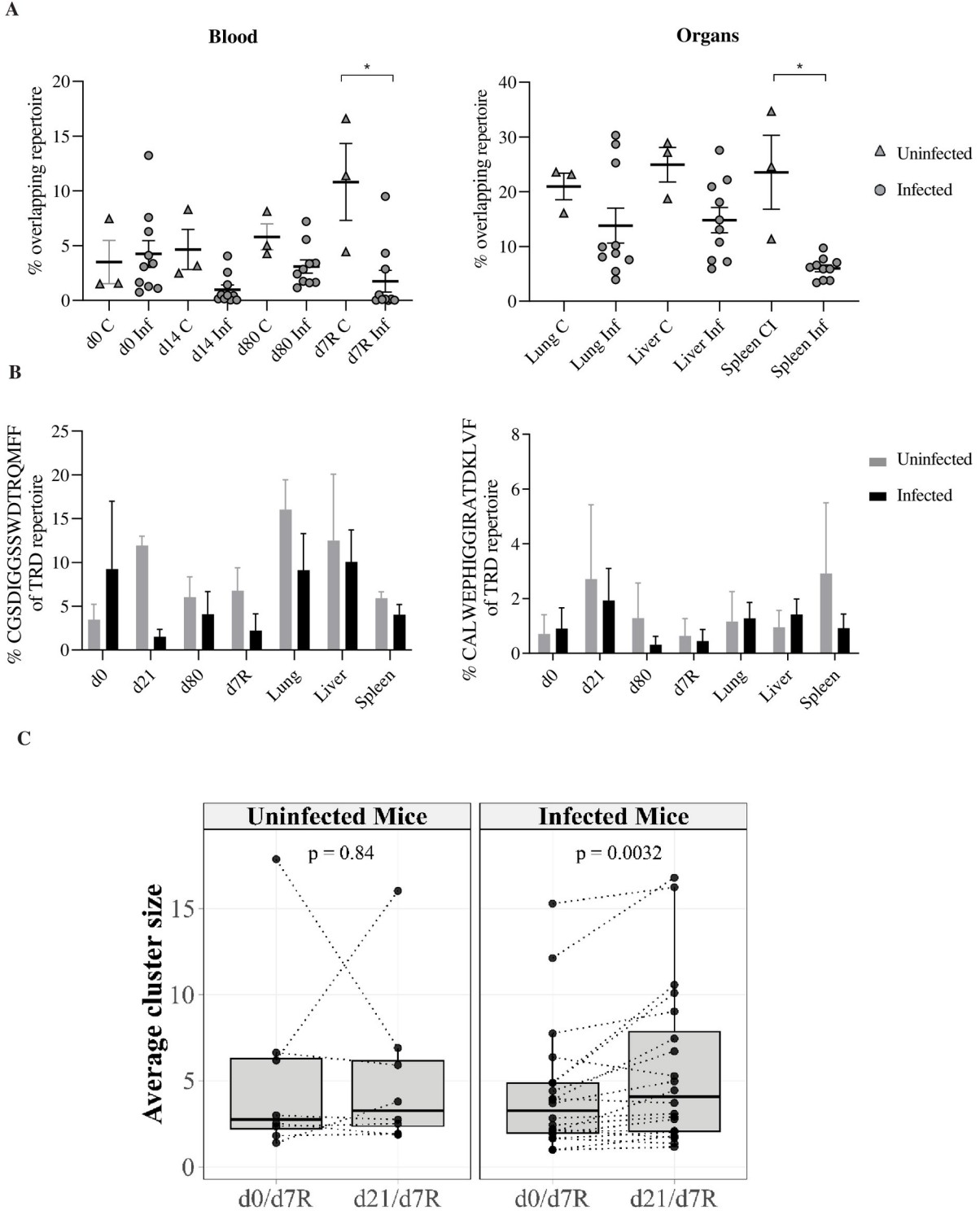

**Fig 6. Expansion of private CDR3δ repertoire following MCMV infection.** TCRα-/- mice (n = 5) were infected with MCMV. Age matched uninfected TCRα-/- mice (n = 3) injected with medium were used as controls. Blood was drawn at different time intervals post-infection and at d7 post-reinfection. (A) Geometric mean of relative overlap frequencies (F metrics) within pairs of blood (left) or of organ (right) samples, each dot represents the F value (x100) of a pair of samples. Statistical tests were Mann Whitney. (B) Percentage of CGSDIGGSSWDTRQMFF (left), and CALWEPHIGGIRATDKLVF (right) sequences in TRD repertoire, in blood and organs from infected and naïve age matched control mice. Statistical test was 1-way ANOVA. (C) CDR3aa clustering of d0 and d21 per mouse. For each mouse, we calculate the Levenshtein distance of d0 and d21 repertoire with d7R reinfected repertoire (liver, lung, spleen). For each comparison (d0/d7R and d21/d7R), we evaluate the average cluster size and perform a paired Wilcoxon test.

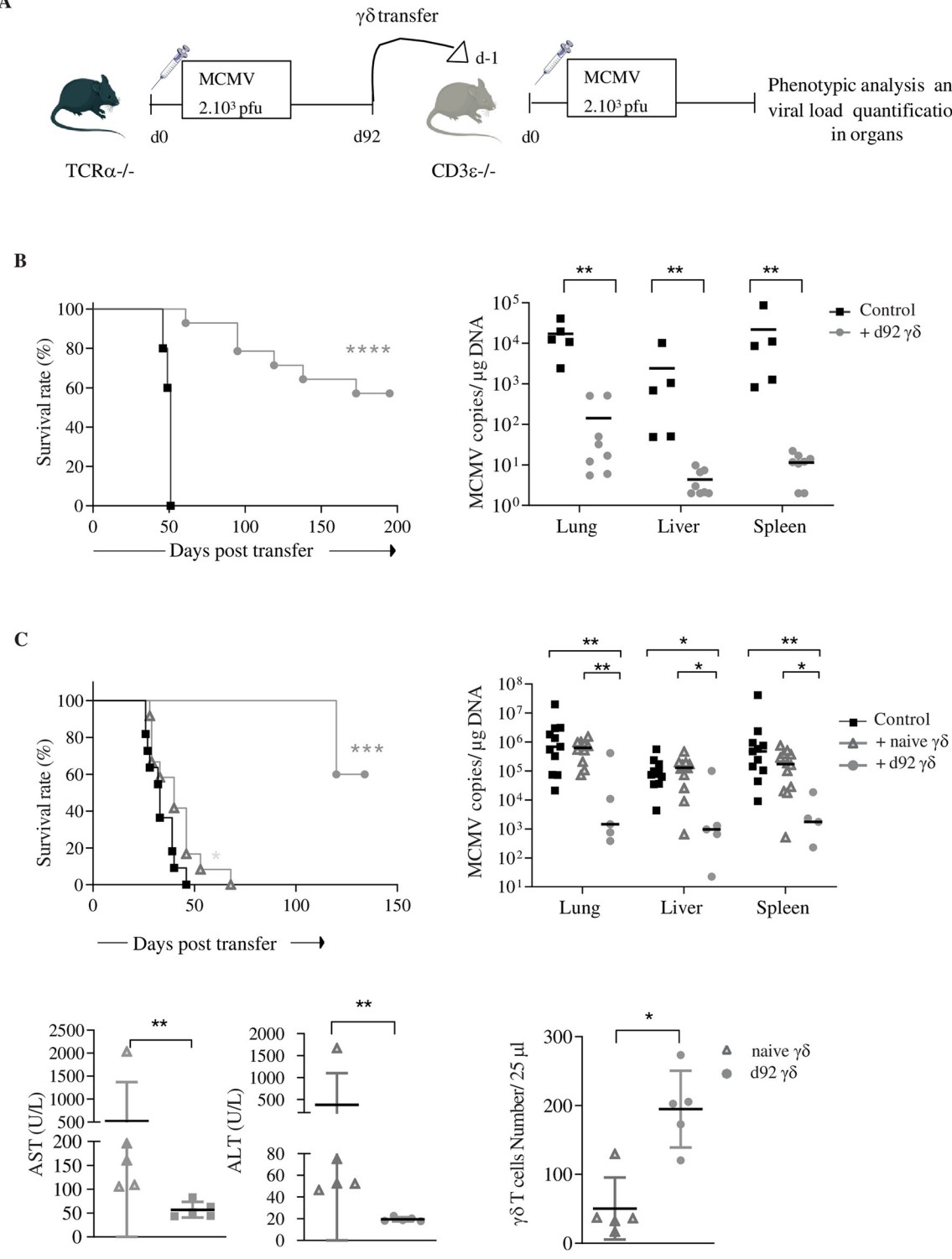

**Fig 7. Long-term induced γδ T cells confer protection to T cell deficient hosts upon adoptive transfer.** (A) Experimental scheme. γδ T cells were sorted from the spleen (purity > 95%) of 3 months infected TCRα-/- mice and $1.10^6$ cells were transferred into CD3ε-/- mice. The day after, CD3ε-/- hosts were infected with MCMV ($2.10^3$ PFU). (B, left) Survival curve of mice given long-term MCMV-primed γδ T cells (grey line, n = 14) and of CD3ε-/- control mice (dark line, n = 5). Mice were sacrificed when losing 10% weight. (B, right) Quantification of viral loads by qPCR, in organs from 5 T-cell deficient mice at sacrifice (dark squares) and 8 γδ-bearing mice that

had survived until d130 (grey circles). (C) $1.10^6$ γδ T cells from age-matched TCRα-/- control mice were transferred into CD3ε-/- mice that were MCMV-infected the day after. (C, upper left) Survival curve of mice given unprimed (naïve) γδ (n = 12), MCMV-primed γδ (n = 5), and of untransferred CD3ε-/- mice (n = 11). (C, upper right) Quantification of viral loads by qPCR, in organs from CD3ε-/- control mice (n = 11), naïve γδ T cell bearing mice (n = 11), and MCMV-primed γδ T cell bearing mice (n = 3) (C, lower panels) Serum quantification of aspartate aminotransferase (ALT) and alanine aminotransferase AST (left) and number of γδ T cells in blood (right) at day 30 post transfer, in 5 mice receiving unprimed (naïve) or MCMV-primed γδ T cells. Statistical tests were Mann Whitney. Log-rank test were used for Kaplan-Meier survival curves. Survival curves and quantification of viral loads were from one representative experiment out of two. Images of mice and syringes were drawn by using pictures from Servier Medical Art. Servier Medical Art by Servier is licensed under a Creative Commons Attribution 3.0 Unported License (https://creativecommons.org/licenses/by/3.0/).

Thus, long-term imprinting by MCMV endows γδ T cells with an enhanced protective potential which may (at least partially) depend on a longer half-life in T-cell deficient hosts.

## 8. The protective function of MCMV-induced γδ T cells principally relies on the presence of γδ TCM

Long-lasting memory has long been attributed to αβ TCM, while the function of γδ TCM remains poorly studied. To gain knowledge on the protective function of these memory γδ T cell subsets, we sorted γδ TCM, TEM KLRG1+ and TEM KLRG1- from the spleen of d92 MCMV-infected TCRα$^{-/-}$ mice, and their protective function after transfer was assessed as above. Interestingly, γδ TCM was the only subtype able to confer good protection to T-cell deficient hosts upon adoptive transfer (Fig 8A, upper panel). In moribund mice from the TEM KLRG1+ and KLRG1- transfer groups, viral loads (Fig 8A, lower panels and Tab H in S1 Data) and transaminase levels (Fig 8B) were elevated, with no statistical difference comparatively to the non-transferred group. In contrast, for both readouts, significant differences were observed between CD3ε$^{-/-}$ control mice and surviving mice from the TCM transfer group (Fig 8A, lower panels, Fig 8B and Tab H in S1 Data). Markedly, the majority of γδ T cells recovered in γδ TCM recipient mice display a CD44+CD62L- TEM phenotype, and were for the most composed of KLRG1+CX3CR1+ cells in lung, and KLRG1-CX3CR1- cells in the liver and spleen (Fig 8C and Tab H in S1 Data).

Long-lasting memory relies on the capacity of αβ TCM to survive in the absence of antigen. To test the ability of γδ T cell memory subsets to survive in MCMV-naïve hosts, γδ TCM and TEM subtypes were sorted from three months-infected TCRα$^{-/-}$ mice, and transferred into uninfected CD3ε$^{-/-}$ mice (S10A Fig, top). A much lower proportion of γδ T was evidenced in the blood three weeks after transfer of TEM as opposed to TCM (S10A Fig, lower left and Tab H in S2 Data), revealing their low capacity to survive in the absence of CMV. In this novel setting, only CD3ε$^{-/-}$ recipients that had received γδ TCM three weeks earlier were able to fight MCMV infection (S10A Fig, lower right). Consistently, when cultured *in vitro* in the presence of IL-15, γδ TCM from d92-infected mice exhibited higher proliferative potential comparatively to γδ TEM (S10B Fig and Tab H in S2 Data).

These results univocally show that γδ TCM are precursors of KLRG1+ and KLRG1- γδ TEM. Low numbers of MCMV-primed γδ TCM are sufficient to maintain long-term antiviral activity against MCMV in T-cell deficient hosts, probably through their capacity to survive and to generate γδ TEM able to control viral loads in target organs.

## 9. Single cell profiling reveals distinct characteristics of γδ TCM and TEM responding to infection

To gain further insight in the molecular characteristics of memory anti-viral γδ T cells, we carried out multiome (gene expression and accessibility) single cell analysis of splenic γδ T cells

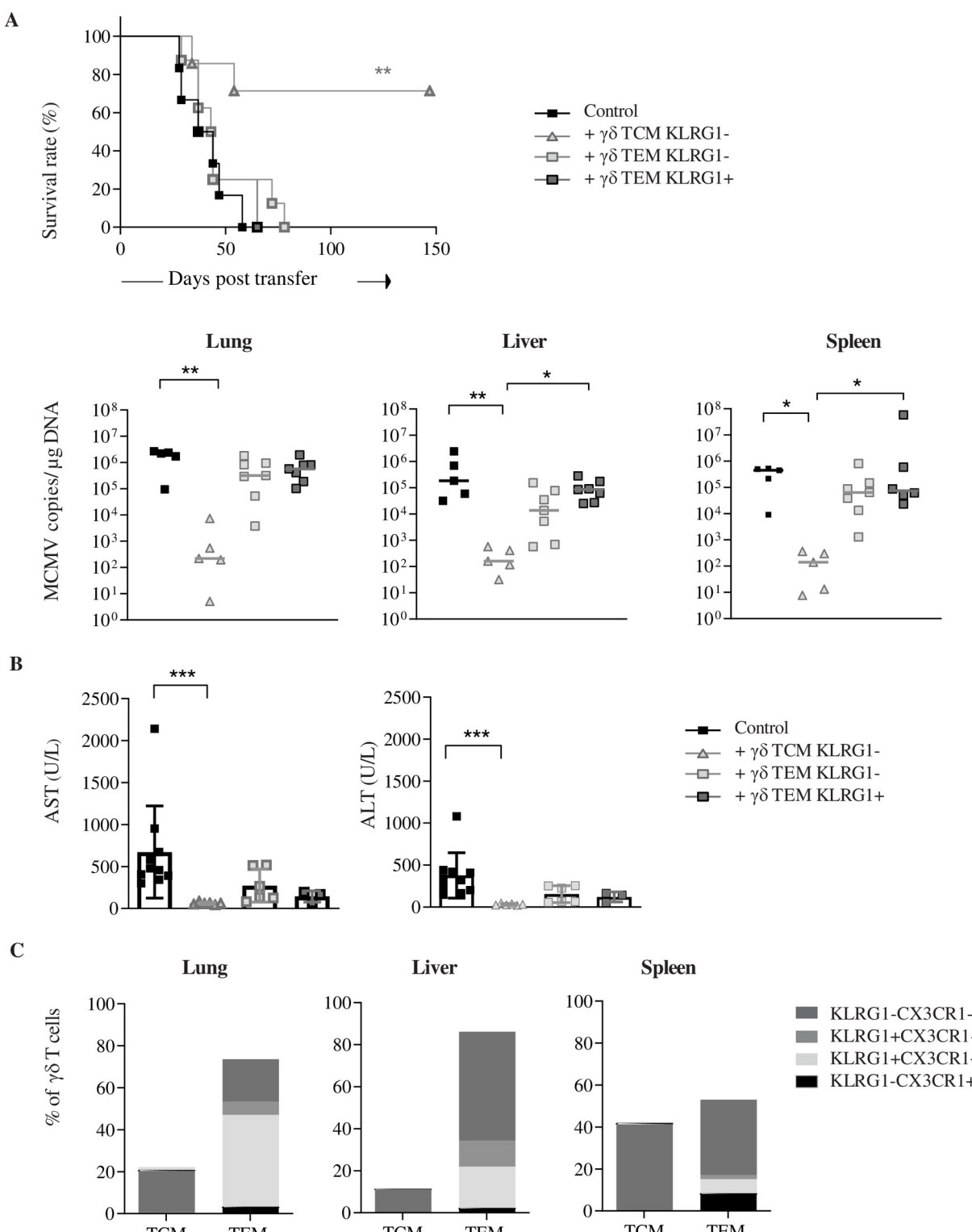

**Fig 8. The protective function of d92 MCMV-induced γδ T cells from the spleen principally relies on the presence of γδ TCM.** (A, upper panel) Survival curve of CD3ε-/- mice given 200000 γδ TCM KLRG1- (n = 7), γδ TEM KLRG1- (n = 8) or γδ TEM KLRG1+ (n = 8). The experiment was repeated twice with comparative results. (A, lower panel) Quantification of viral loads by qPCR in organs from recipient mice at sacrifice. (B) Serum quantification of ALT and AST. Mice given γδ TEM KLRG1- and KLRG1+ cells were sacrificed when losing 10% weight, and mice given γδ TCM KLRG1- were sacrificed at the end of the experiment, at d150 (n = 5). (C) Distribution of γδ T cell memory subsets in organs from γδ TCM recipient mice that had survived. Same statistical test as in Fig 7.

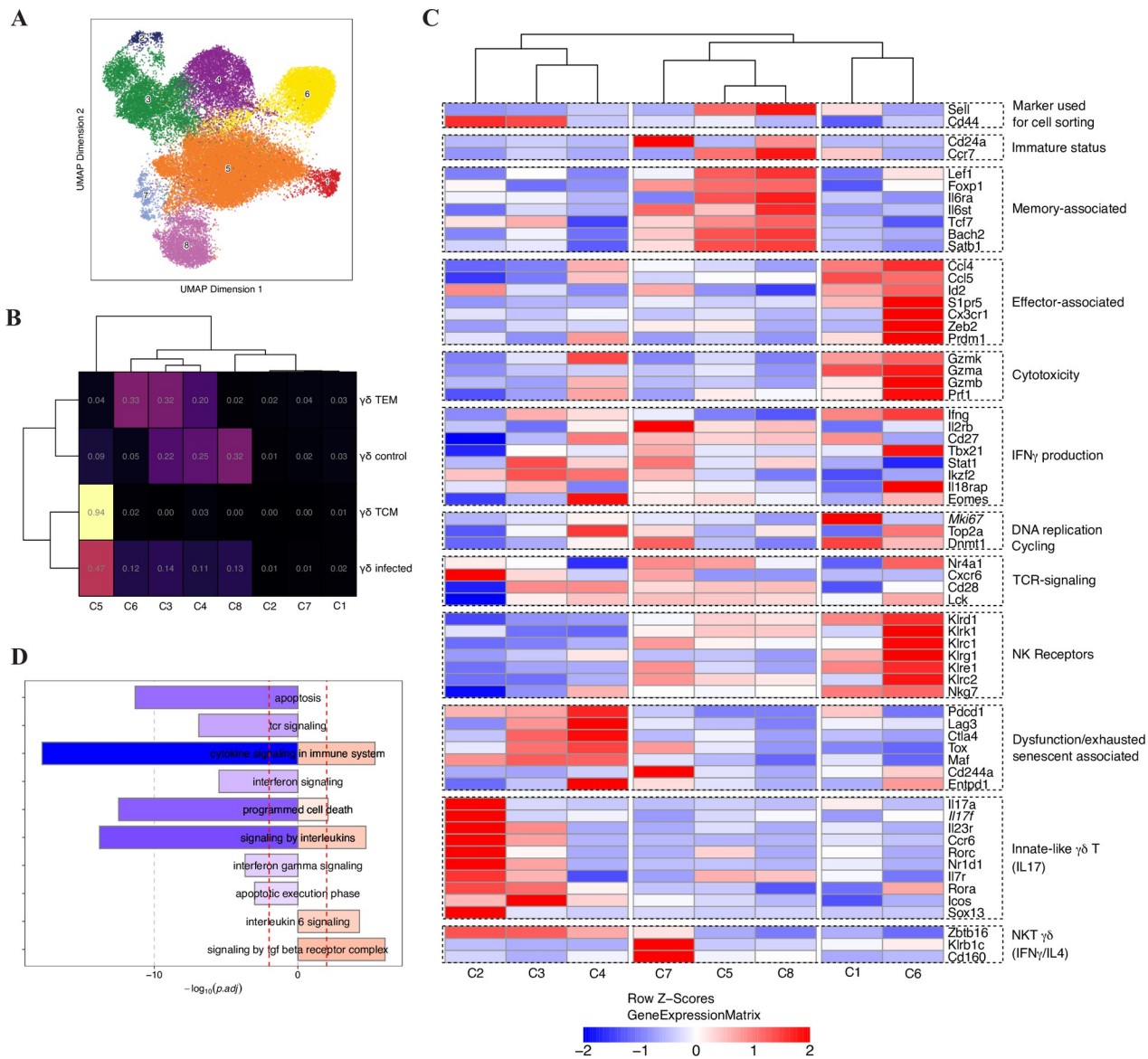

**Fig 9. Single cell profiling reveals distinct characteristics of γδ TCM and TEM responding to infection.** γδ T cells were isolated from pooled splenocytes of 3-months MCMV-infected TCRα-/- mice (n = 15) (γδ infected), or from aged-matched uninfected mice (n = 9) (γδ control). γδ TCM and TEM were then sorted from pooled γδ T cells of infected mice. (A) UMAP projection of 38,819 scATAC-seq and scRNA-seq profiles of γδ lymphocytes combining 4 samples with dots representing individual cells and colors indicating independent clusters. (B) The heatmap represents the cluster confusion matrix for each sample. The values indicate the proportion of cells in each cluster, calculated for each sample (row). (C) The heatmap shows the scaled of the mean log-normalized gene expression across cells among clusters (column) for a list of selected genes (row), categorized by 'functionality'. Genes are significant (FDR ≤ 0.05), except for Mki67 and Il17f, which are italicized. Column clustering was performed using Euclidean distance and the Ward.D2 method. Row Z-scores are constrained to an upper bound of 2 and a lower bound of -2. (D) Bar plots show a list of selected pathways from the Reactome database that are significantly enriched for up- and down-regulated genes from the comparison of C5 and C6 clusters. Pathways are presented using -log10 (p.adj), with pathways from down-regulated genes in C5 vs. C6 shown as negative values on the left (in blue) and pathways from up-regulated genes in C5 shown as positive values on the right (in red). The dashed red line indicates p. adj = 0.01.

(whole population) from control and long-term infected mice. γδ T cells from MCMV-primed mice were further on sorted into TCM-like and TEM-like memory subsets and analyzed concomitantly. UMAP visualization of the four individual samples is depicted in S11A Fig (in

red). Eight independent clusters were defined (Fig 9A). S11B Fig and S2 Table display the top 10 genes expressed per cluster, while Fig 9B shows the proportion of each sample in the different clusters. In contrast to γδ TEM encompassing cluster 3 (C3), C4 and C6, γδ TCM formed a compact and distinct group that fell into C5. Few γδ T cells from uninfected mice were present in C5 (9%), and C6 (5%) (Fig 9B), suggesting that the majority of memory cells within these clusters display functional specificity toward CMV.

To get closer to the function and differentiation stage of cells within these two infection-related clusters, we analyzed the relative abundance of memory- and effector-associated markers, based on previous work on αβ CD8 T cells responding to a viral challenge [49–58]. In addition to *Klrg1* and *GzmA* (among the top 10 genes in C6, S11B Fig and S2 Table), γδ T cells from C6 overexpressed effector associated markers such as *Cx3cr1*, *Zeb2*, as well as transcripts related to cytotoxicity *(prf1, Gzmb)*, IFNγ production *(Tbx21, IL18rap)*, and TCR signaling *(Nr4a1)* (Fig 9C and S3 Table for gene expression; S11C Fig and S4 Table for gene accessibility). In contrast, mean expression of emblematic markers of "exhausted" cells (*Pdcd1*, *Ctla4*, *Lag3*, *Tox*) was low in C6 (Figs 9C and S11C), accordingly to previous findings on inflationary αβ CD8 T responding to CMV. On the other hand, cells within C5 and C8 showed increased expression of memory-associated factors/receptors (*Tcf7*, *Lef1*, *IL6Ra*. . .). However, C8 was free of cells from the γδ TCM/TEM infected samples (Fig 9B), indicating the prevalence of immature, MCMV-unprimed γδ T lymphocytes in this cluster. Interestingly, *Itga4* (encoding CD49d, an integrin that was associated to Ag-specific activation for αβ CD8 T cells [59], was among the top 15 genes overexpressed in C5 *versus* C8 (S11D Fig and S5 Table). Similar analysis between C5 and C6 disclosed up regulation of the pro-survival activators *Bcl2* (S11E Fig and S6 Table) in γδ TCM comparatively to TEM, consistently with their better survival after adoptive transfer (S10 Fig). Fig 9D (see also S7 Table) expand the investigation to the pathway level and highlight TGF β- and IL6-signaling as regulatory pathways for γδ TCM integrity, while apoptosis and TCR-signaling appeared to be increased in γδ TEM responding to infection.

Finally, based on previous knowledge on γδ T cells [60–64], we looked for typical signatures of so-called "innate-like" γδ T cells. This analysis prompted us to envision the presence of CD44hi CCR6+ IL17-committed γδ T cells in C2, and NK1.1+ IFNγ/IL-4-committed γδ T cells in C7 (Fig 9C), both unaffected by infection, suggesting their absence of response to CMV.

This detailed scrutiny underlines shared characteristics between αβ and γδ T cell memory subsets, and suggest differentiation of immature γδ T cells (C8) toward primed γδ TCM (C5) and TEM (C6) in response to CMV encounter, thus sustaining the adaptive-like features of γδ T cells in this context.

## Discussion

In contrast to increasing numbers of studies showing the memory potential of γδ T cells during acute resolving infections, data describing γδ T cell memory in persistent infections are missing. CMV is the prototypical β-herpesvirus establishing lifelong latent infection. Anti-CMV memory responses are essential to control viral reactivation and/or reinfection events that commonly take place in SOT. The role of γδ T cells could be significant in this context where immunosuppressive/ablative treatments cause suboptimal and/or delayed αβ T cell responses [22]. In the present study, we used subsequent MCMV infections of TCRα-/- mice to decipher the memory potential of γδ T cells against CMV. Our data depict a more rapid γδ T cell secondary *versus* primary antiviral response in MCMV-target organs and blood, with the implication of γδ TEM. They are in line with the faster increase of non-Vγ9Vδ2 TEMRA cells

in blood from secondary infected HCMV-seropositive *versus* primary infected HCMV-seronegative renal transplant recipients, which associates to shorter infection resolution [26,29]. Nonetheless, in reference to past knowledge, the extent of the secondary γδ T cell response to CMV appears less pronounced than that of CMV peptide-specific αβ CD8 T cells, that can be easily followed by the use of tetramers. This is barely comparable to the response of a whole population of γδ T cells, in which we do not now the proportion of cells responding to the virus.

Stressing the value of our mouse model, our adoptive transfer experiments show a major role for γδ TCM in the maintenance of long-term antiviral activity. Furthermore, our study points out that γδ T cell memory function can operate independently of priming with αβ CD4 T cells. However, DNA copies of MCMV were still detected in organs three months after infection (Figs 3B and 5C). Thus, γδ T cells are probably not sufficient to fully control the virus in the absence of αβ T cells. Despite this, MCMV drove the accumulation of γδ TEM that were mostly PD-1[neg], unlike T cells responding to chronic viruses such as non-cytopathic LCMV clone 13 [7]. In fact, γδ TEM from d92-infected mice expressed KLRG1, a typical feature of CMV-induced αβ CD8 TEM that was associated to antigen persistence rather than to viral replication [65,66]. These results emphasize the importance of the intrinsic nature of pathogen in determining T cell fate, although the route and dose of infection may also operate [67,68].

We show here mutually exclusive expression of KLRG1 and PD-1 on γδ TEM, related to their anatomical localization. KLRG1+ γδ TEM co-expressed CX3CR1 and were mainly located in the blood and intravascular compartment from organs, while PD-1+ γδ TEM prevailed beyond the vasculature. Our results are in line with the recent study by Oxenius's team, who depicted a rise of pulmonary intravascular (IV+) KLRG1+ αβ CD8 TEM driven by MCMV [69]. Interestingly, the increase of KLRG1+ γδ TEM in the intravascular compartment occurred concomitantly to a decrease of KLRG1- γδ T cells in the IV- fraction, suggesting a possible modulation of this receptor while γδ T cells transit from the parenchyma to the blood circulation, and reciprocally. The accumulation of KLRG1+ γδ TEM in long-term infected mice might thus reflect their continuous dissemination through the bloodstream in response to systemic MCMV infection. As shown earlier for "inflationary" αβ CD8 T cells, γδ TEM from the liver and lung of long-term MCMV-infected mice were poorly affected by the FTY-treatment, suggesting that their maintenance does not depend on antigen encounter within the lymph nodes [35].

Priming of γδ T cells with MCMV induced transcription of genes associated to cytotoxicity, in accord with their reported capacity to kill CMV-infected fibroblasts, a function that could be important to control the virus *in vivo* [28]. Long-term MCMV-primed γδ T cells also upregulated genes encoding NKR, including signaling lymphocyte activation molecules (SLAM). The latter are commonly found on cytotoxic effectors and may act as important rheostats to fine-tune their functions. The modulation of these transcripts along the course of MCMV infection and reinfection is reminiscent to repetitive antigen exposure [70], suggesting continuous triggering of the γδ TCR by still-unknown MCMV-induced antigens. The concomitant downregulation of genes/protein involved in trafficking into lymph nodes suggest a preferential function of these cells in the tissues such as the liver and lung.

We showed the capacity of long-term MCMV-primed γδ T cells to confer protection upon adoptive transfer into T-cell deficient hosts; in contrast, the time-delay needed to proliferate and differentiate probably precludes MCMV-naïve γδ T cells to counteract the amplification and dissemination of viral copies.

First evidence for γδ T cell memory responses were put forward in the early 2000 and implicated phosphoantigen-reactive Vγ9Vδ2 in primates [71–73]. In mice, pioneering work was carried out by Lefrançois's team who showed higher increase of Vγ6Vδ1 cells in mesenteric

lymph nodes after *Listeria monocytogenes* (*Lm*) secondary and even tertiary oral challenge, relatively to primary infection [74,75]. Since then, increasing numbers of studies have shown the memory potential of γδ T cells in infections ([41,42,44] and reviewed in [13–15]). The γδ T cell subtypes described in most of these studies are pre-activated lymphocytes with innate-like features and limited TCR diversity. In contrast, a tendency towards an increase in TRD diversity could be noticed in blood after MCMV reinfection, concomitant to a decrease of the highly frequent innate like CGSDIGGSSWDTRQMFF sequence, and to the acquisition of a more private TRD repertoire. Since innate-like γδ T cells are commonly shared between mice, our results suggest the mobilization/amplification of private adaptive-like γδ T cell clones soon after MCMV encounter. They are in line with our previous data showing, by the use of CD3ε$^{-/-}$ mice receiving TCRα$^{-/-}$ bone marrow graft, that fetal-derived γδ T cells are dispensable for long-term MCMV protection in the adult mice [27]. The participation of innate-like γδ T cells in the antiviral response remains possible, and might be particularly relevant in humans where congenital HCMV infection can occur [76,77]. Interestingly, expansion of private CDR3β clones at the expense of more public ones was described by Friedman and colleagues following immunization with either self or non-self MHC-restricted peptides [47]. The implication of adaptive-like γδ T lymphocytes in the antiviral response is supported by the rise of the γδ TEM/TCM ratio along the course of MCMV infection. Likewise, HCMV infection induces the differentiation of non-Vγ9Vδ2 T cells from a naïve to TEMRA phenotype [29,78]. However, in contrast to the studies in humans which describe HCMV as a major driver of TRD repertoire focusing over time in transplant patients [22,32,79] the overall TRD diversity in blood from long-term infected mice (d80) was similar to that of control mice. This suggests that the TRD repertoire somehow stabilizes at distance from MCMV challenge, likely due to the loss of some adaptive-like γδ T cell effectors, and to the high frequency of innate like γδ TEM in mice regardless of infection. Alternatively, the transplant setting in the HCMV studies may contribute to the high expansion of particular γδ clonotypes.

Our TCRγδ-blocking experiment suggest that TCR-signaling is important for viral load control after reinfection. However, we cannot fully exclude that γδ TCR-antibody also affects effector function or survival of innate or cytokine receptor-dependent activation of γδ T cells. Whether γδ T cell memory response involves TCR binding of diverse MCMV-induced antigens remains to be elucidated [16,80]. Counterintuitively, γδ clonotypes that dominated the chronic phase of infection (d80) were not necessarily expanded in the initial phase of infection (d21). Likewise, expansion of γδ T cell clonotypes after first MCMV challenge was not a prerequisite for their involvement in the secondary response. In an elegant study, the group of Buchholz recently showed that, during MCMV infection, long-term immunodominance of a single-cell derived αβ T cell family is not determined by its initial expansion, but is rather predicted by its early content of T central memory precursors [81]. Moreover, inflationary KLRG1+ αβ CD8 T cells have a reduced half-life and are supposed to be maintained by continuous replenishment from early primed KLRG1- αβ T cells, among which αβ TCM expressing Tcf1 [31,51,82–84]. We hypothesize that γδ TCM primed by a first MCMV encounter play a crucial role in the long-term anti-CMV response, through their capacity to survive/proliferate and to give rise to antiviral effectors that eventually localize in target tissues or recirculate. In accordance with our hypothesis, we showed up-regulation of *Bcl2* in TCM *vs* TEM, the high proliferative potential of γδ TCM *in vitro*, as well as their protective function and their ability to generate KLRG1+ γδ TEM upon adoptive transfer into T cell deficient hosts. Our transcriptomic and epigenomic results also show a close homology of γδ TCM with αβ TCM supported by *Sell*, *Ccr7*, *Il7r* and *Tcf7* gene expression. Clear differences between γδ TCM and γδ TEM after long-term MCMV infection were also evinced, that are consistent with a better survival of γδ TCM, and superior cytotoxic effector functions and exhausted features of γδ TEM.

Applying a second intraperitoneal injection of MCMV to primarily infected mice probably leads to the mobilization of both novel γδ T cell responders and of preexisting γδ TCM, thus explaining the appearance of novel γδ TCR clonotypes, and to an enhanced γδ T cell secondary response.

To conclude, our results uncover memory properties of γδ TCM cells in the context of a chronic viral infection. These data reveal the interest of investigating and targeting this subset of unconventional T cells in strategies aiming at improving antiviral vaccination approaches. γδ TCM could compensate for the defect of αβ T cells in immunosuppressed individuals and are thus of particular relevance in the context of organ or hematopoietic stem cell transplantation.

## Methods

### Ethic statement

Animal experiments were carried out in accordance with the Ethics Review Committee of Bordeaux and were approved by the French Ministry of Higher Education and Scientific Research (approval number APAFIS#7022–2016092917471799).

### Mice

Specific Pathogen-free, 8–12 weeks old, C57BL/6 $CD3\varepsilon^{-/-}$[85] as well as $TCR\alpha^{-/-}$ [86] were purchased from the CDTA (Centre de Distribution, Typage et Archivage Animal, Orléans, France). Experiments were performed in an appropriate biosafety level 2 facility in compliance with governmental and institutional guidelines (Animalerie spécialisée A2, Université Bordeaux Segalen, France, approval number B33-063-916).

### Virus

MCMV (Smith strain, ATCC VR-194) was obtained from the American Type Culture Collection and propagated into BALBc mice (BALBcBy/J, Charles Rivers laboratory, Larbresle, France) as previously described. Three weeks after MCMV infection, salivary glands were collected from infected mice and used as an MCMV stock solution. Determination of virus titers was defined by standard plaque forming assay on monolayers of mouse embryonic fibroblasts (MEF). Infections were performed by intraperitoneal (i.p.) administration of desired quantity of PFU from the salivary gland viral stock.

### Flow cytometry

For lung preparation, collagenase I (50 μg/ml; Sigma) and DNase I (50 μg/ml; Sigma) were used for 1h at 37˚C. Single-cell suspension from spleen, liver and enzyme treated lung were prepared by meshing organs through 70 μm nylon cell strainers in RPMI-1640 with 8% FBS (HyClone Laboratories, GE Healthcare, Logan, Utah). After red blood cells lysis with NH4Cl (liver) or ACK (lung), lymphocytes were isolated by a discontinuous 40/80% Percoll gradient (GE Healthcare).

For Flow cutometry analysis, before antibody staining, Fc-receptors were blocked with CD16/CD32 FcR antibody (clone 93, eBioscience). Live/dead discrimination was performed using Fixable Viability Stain 700 (BD Horizon) according to manufacturer's instructions. The following monoclonal antibodies were used:

For detection of intracellular cytokines, we employed the BD Cytofix/Cytoperm fiixation/Permeablization kit. Cells were acquired using a LSRFortessa (BD Biosciences), and analyzed using FlowJo software (Tree Star).

| Antibody | Fluorochrome | Clone | Concentration | Supplier |
|---|---|---|---|---|
| γδ | BV421 | GL3 | 0,2 mg/ml | BD Biosciences |
| CD19 | BV421 | 6D5 | 0,2 mg/ml | Biolegend |
| CD3e | BV786 | 145-2C11 | 0,2 mg/ml | BD Biosciences |
| CD44 | BV500 | IM7 | 0,2 mg/ml | BD Biosciences |
| CD62L | PerCP-Cy5.5 | MEL-14 | 0,2 mg/ml | BD Biosciences |
| Cx3CR1 | PE-Cy7 | SA011F11 | 0,2 mg/ml | Biolegend |
| Cx3CR1 | PE | SA011F11 | 0,2 mg/ml | Biolegend |
| Granzyme A | PE | 3G8.5 | 0,2 mg/ml | Biolegend |
| PD1 | FITC | 29F.1A12 | 0,5 mg/ml | Biolegend |
| KLRG1 | APC | 2F1 | 0,2 mg/ml | BD Biosciences |
| KLRG1 | FITC | 2F1 | 0,5 mg/ml | eBioscience |

### Anti-CD45 intravascular staining

For in vivo antibody labelling, a total of 10 μg/100 μl of anti-CD45 APC (clone 30-F11 from eBioscience; 0,2 mg/ml) was injected i.v to anesthetized mice via the retro-orbital venous plexus as previously described [87]. Antibody was allowed to circulate for 5 min to label cell in circulation or closely related to the vascular space. After this lapse-time, animals were sacrificed. Spleen, liver and lung were removed. Cells were isolated for ex vivo labelling before flow cytometry.

### FTY720 treatment

To prevent lymphocyte recirculation and egress from lymph nodes, the S1P1R agonist, FTY720 (2- amino-2-[2-(4-octylphenyl)ethyl]-1,3-propanediol) was used. FTY720 (Sigma-Aldrich) was reconstituted in ethanol and diluted in 2% β-cyclodextrin (Sigma-Aldrich) for injections. Long-term MCMV infected TCRα$^{-/-}$ mice (d92 post primary infection) received every other day i.p. injections of FTY720 at a dose of 1 mg/kg, or of vehicle control containing 2.5% ethanol and 2% β-cyclodextrin. FTY720 was administered for a period of 11 days. Cells from spleen, liver and lung were analyzed one day after the final FTY720 injection.

### Multiplex gene expression analysis

γδ T cells were sorted from the lung and liver of TCRα$^{-/-}$ mice (n = 10), using the TCR γδ T cell Isolation kit (Miltenyi Biotec). RNA was extracted using the Nucleospin RNAII kit (Macherey Nagel), was quantified with a NanoDrop spectrophotometer (Promega), and was qualified with the 2100 Bioanalyzer System (Agilent). The nCounter GX analysis system (NanoString) was utilized according to the manufacturer's directions to quantify RNA expression of 547 genes on the nCounter Immunology Panel.

### DNA extraction and quantification of MCMV DNA copy number

Genomic DNA from organs was isolated using a Nucleospin tissue purification kit (Macherey Nagel). MCMV-DNA was quantified by real-time PCR using BIO-RAD CFX with GoTaq qPCR Master Mix (Promega) and primers specific for MCMV glycoprotein B (gB) (forward primer: GGTAAGGCGTGGACTAGCGAT and reverse primer: CTAGCTGTTTTAACGC GCGG). Samples were distributed by Eppendorf epMotion 5073 automated pipetting. PCR condition was as follows: 95°C for 10 min, denaturation at 95°C for 15 s, and annealing/extension at 60°C for 1 min. Known quantities of plasmid comprising MCMV gB were used for the titration curve.

## In vivo treatment with anti- γδ TCR mAb

TCRα$^{-/-}$ mice received 200 μg/mouse of purified Hamster anti-mouse γδ T-Cell Receptor (Clone GL3) mAb, or Hamster IgG2, κ Isotype Control (Clone B81).

## CDR3 TCRδ (TRD) high-throughput sequencing

RNA was prepared from blood (250 μl) of individual mice using QIAmp RNA Blood kit (Qiagen) and NucleoSpin RNA blood kit (Macherey Nagel), or from sorted gd T cells (organs) as above. cDNA was generated performing a template switch anchored RT-PCR. RNA was reverse transcribed via a template-switch cDNA reaction using 5' CDS oligo (dT), a template-switch adaptor (5'-AAGCAGTGGTATCAACGCAGAGTACATrGrGrG) and the Superscript II RT enzyme (Invitrogen). The cDNA was then purified using AMPure XP Beads (Agencourt). Amplification of the TRD region was achieved using a specific TRDC primer (5'- *GT CTCGTGGGCTCGG*AGATGTGTATAAGAGACAGAAAACAGATGGTTTGGCCGGA, adapter in italic) and a primer complementary to the template-switch adapter (5'- *TCGTCGG CAGCGTCAGATGTGTATAAGAGACAG*AAGCAGTGGTATCAACGCAG, adapter in italic) with the KAPA Real-Time Library Amplification Kit (Kapa Biosystems). Adapters were required for subsequent sequencing reactions. After purification with AMPure XP beads, an index PCR with Illumina sequencing adapters was performed using the Nextera XT Index Kit. This second PCR product was again purified with AMPure XP beads. High-throughput sequencing of the generated amplicon products containing the TRG and TRD sequences was performed on an Illumina MiSeq platform using the V2 300 kit, with 150 base pairs (bp) at the 3' end (read 2) and 150 bp at the 5' end (read 1) [at the GIGA center, University of Liège, Belgium].

Raw sequencing reads from fastq files (read 1 and read 2) were aligned to reference V, D and J genes from GenBank database specifically for 'TRD' to build CDR3 sequences using the MiXCR software version 3.0.13 [88]. Default parameters were used except to assemble TRDD gene segment where 3 instead of 5 consecutive nucleotides were applied as assemble parameter. CDR3 sequences were then exported and analyzed using VDJtools software version 1.2.1 using default settings [89]. Sequences out of frame and containing stop codons were excluded from the analysis. Note that the nucleotype lengths generated by VDJtools include the C and V ends of the CDR3 clonotypes. The degree of TCR repertoire overlap between two different samples was analyzed using the overlap F metrics calculated with the software package VDJtools (https://vdjtools-doc.readthedocs.io/en/master/index.html).

To evaluate the bias in the γδ TCR repertoires [90], we computed a distance matrix of pairwise Levenshtein distances (LD) between CDR3aas, using 'stringdist' R package (M.P.J. van der Loo, 2014, The R Journal 6(1):111–122). When two sequences are similar under the defined threshold, LD>1 (i.e., at most one amino acid difference), they were connected and designated as "cluster". CDR3s with more than one amino acid difference from any other sequences are not connected. The number of CDR3aa making up the cluster represents the size of the cluster. The clustering was between d0 and d7R (d0/d7R) and between d21 and d7R (d21/d7R). The mean size of the clusters was compared between d0/d7R and d21/d7R per mouse.

## Adoptive transfer of sorted γδ T cells

γδ T cells were sorted from the spleen of MCMV infected or control TCRα$^{-/-}$ mice using the TCR γδ T cell Isolation kit (Miltenyi Biotec). For γδ TCM and TEM sorting, purified γδ T cells were stained and isolated using FACSAria II Sorter (BD Biosciences). Purity was > 95%. Recipient CD3ε$^{-/-}$ mice received intravenous (i.v.) injections of a defined number of total γδ T

cells or of memory γδ T cell populations. These animals were infected with $2.10^3$ pfu of MCMV the day after and followed twice a week. Mice were sacrificed when losing 10% of their original weight or when showing signs of distress.

## AST and ALT quantifications

Mice were bled via the retro-orbital sinus after anesthesia. Serums were collected and frozen before quantification using a clinical chemistry analyzer (Horiba Pentra C400).

## Preparation of single cell libraries

TCRα$^{-/-}$ mice (n = 15) were infected with MCMV for 3 months. Splenocytes were pooled and γδ T lymphocytes isolated using the TCR γδ T cell Isolation kit (Miltenyi Biotec). γδ TCM and TEM from infected mice were stained and sorted using FACSAria II Sorter (BD Biosciences). Purity was > 95%. Whole γδ T cells were also prepared from the spleen of age-matched TCRα$^{-/-}$ mice (n = 9). Nuclei were isolated following the 10x Genomics protocol for use with the Chromium Next GEM Single Cell Multiome ATAC + Gene Expression (GEX) protocol. 4 libraries were prepared for scRNASeq (1 = γδ TCM, 2 = γδ TEM, 3 = γδ infected and 4 = γδ control) and 4 for scATACSeq (5 = γδ TCM, 6 = γδ TEM, 7 = γδ infected and 8 = γδ control). Sequencing was performed by IntegraGen (Evry, France) using Novasesq 6000 2x100bp.

## Single cell analysis

**10x Genomic Cellranger arc count** (v2.0.2) was used to process in parallel scATAC-Seq and scRNA-Seq fastq data. The 10X Genomic pre-built reference *refdata-cellranger-arc-mm10-2020-A-2.0.0* was used for alignment. The R add-on package ArchR (1.0.1) was used for downstream analysis. scATAC-Seq count data were imported into R (v4.3.0) as ArrowFiles using the function **createArrowFiles**. Arrow files were combined into an ArchProject using the function **ArchRProject** which produced a "Tile Matrix", count of reads across genome-wide 500-bp bins. scRNA-Seq count data were imported into R using the function **import10xFeatureMatrix** and added to the ArchProject using the function **addGeneExpressionMatrix**. TSS positions were acquired from the *BSgenome.Mmusculus.UCSC.mm10*, Bioconductor package. Cells were filtered out if (i) there are less than 2000 fragments detected, (ii) the Transcription Start Site (TSS) Enrichment Score is less than 8 and (iii) without associated transcriptomic data. Potential doublets were identified and removed using the function **addDoubletScores** and **filterDoublets.** Two-round clustering strategy was used for dimensional reduction and cell clustering. The first round is based on the 500-bp tile matrix using iterative LSI (Latent Semantic Indexing) method. This step allows to create pseudo-replicates (clusters of cells) which permit to perform reproducibly peak calling. Peak calling is among the most fundamental processes in ATAC-seq data analysis. For the second round, peaks were called and merged in each pseudo-replicates obtained from the first round using **MACS2**. Thus, a count matrix was constructed by counting the reads related to Tn5 insertion per peak for each cell. Iterative LSI was performed also on scRNA-Seq previously imported as well as on the peak-by-cell matrix. Harmony algorithm was performed on the combined LSI dimensions for batch correction using the function **addHarmony**. Cell clustering was performed on the Harmony dimensions using the function **addClusters** (clustering based on Shared Nearest Neighbors network and Louvain's method for communities detection). During this round of clustering, resolutions of 0.1, 0.2, 0.3, 0.4, 0.5, 0.6, 0.7, 0.8, 1 were generated, and a final resolution of 0.2 was chosen. Uniform Manifold Approximation and Projection (UMAP) embedding was used for representing cells on 2 dimensions (function **addUMAP**). Gene scores matrix was calculated using the function **addGeneScoreMatrix**. This is a prediction of a gene expression based on

the accessibility of regulatory elements in the surrounding of the gene. Cluster markers identification was performed on Peak, Gene score and RNA matrices using the Wilcoxon's test through the function **getMarkerFeatures.** Significant level was set at alpha = 0.01 (type 1 error). Over-representation analysis was performed using the function **fora** in the R package **fgsea** to test for the enrichment of cluster marker gene lists on biological function databases such as Reactome (through the R package **msigdbr**).

Data from single cell analysis are available at GEO repository under accession number GSE269394

## Statistical analysis

Statistical studies were performed using GraphPad Prism 6 software and indicated in the Figure legends (*P <0,05; **P <0.01 ***P <0.001 and ****P <0.0001).

## Supporting information

**S1 Fig.** (A) Co-expression of CX3CR1 and KLRG1 on γδ TEM from organs and blood of d92 MCMV-infected and age-matched control mice. (B) Mutually exclusive expression of KLRG1 and PD1on γδ TEM from organs of d92 MCMV-infected and age-matched control mice. One representative mouse is shown for each (control and long-term MCMV infected) group of mice. (TIF)

**S2 Fig. Multiplex gene expression analysis of immunology markers expressed by MCMV-induced γδ T cells.** γδ T cells were sorted from lungs and liver of TCRα-/- mice (n = 10) at d0, d14 or d92 post-primary MCMV infection, at day 7 post-reinfection, and pooled before RNA extraction. Analyses were performed with nSolver Analysis Software (NanoString). (A) Histograms represent transcripts shared by liver and lungs, and whose d92/d0 ratios were >2 or < -2 (minimum counts = 100). (B) Counts evolution of theses transcripts at day 0, 14, 92 and day 7 post reinfection. (TIF)

**S3 Fig. Analysis of γδ T cell memory through subsequent MCMV infections**: (A, top) Experimental scheme for longitudinal analysis of γδ T cells in blood. TCRα-/- mice (n = 10) were infected with MCMV ($2.10^3$ PFU) at day 0, then re-challenged at day 92 with similar dose of MCMV. Mice were bled at day 0, 1, 3, 7, 14, 21 and 58 post-primary infection and at day 0, 1, 3, 7, 14 and 21 post-secondary infections. (A, bottom) Comparative analysis between primary (1- Inf) and secondary infection (2- Inf) of γδ TEM (left) and γδ TEM KLRG1+ (right). (B) Experimental scheme for analyses in organs. TCRα-/- mice (n = 25) were infected with MCMV ($2.10^3$ PFU) at day 0, or left uninfected (n = 25). Three months later, uninfected mice were primarily infected, and infected mice were re-challenged with similar dose of MCMV. Mice (5) were euthanized at indicated time points for phenotypic analysis and viral load quantification in the liver and lung. Images of mice and syringes were drawn by using pictures from Servier Medical Art. Servier Medical Art by Servier is licensed under a Creative Commons Attribution 3.0 Unported License (https://creativecommons.org/licenses/by/3.0/). (TIF)

**S4 Fig. KLRG1+ γδ T cells dominate the secondary response to MCMV.** (A) Longitudinal analysis of KLRG1+ γδ T cells in blood following the experimental scheme depicted in S3A Fig. Mean absolute numbers +/- SEM of KLRG1+ γδ T lymphocytes in 25 μl of blood are shown. (B) Comparative analysis between primary (1- Inf) and secondary infection (2- Inf) of γδ TEM KLRG1+ numbers in lung and liver (2-way ANOVA). The experiment was repeated

twice with concordant results.
(TIF)

**S5 Fig. CDR3δ repertoire analysis.** TCRα-/- mice (n = 5) were infected with MCMV. Age matched uninfected TCRα-/- mice (n = 3) injected with medium were used as controls. Blood was drawn at different time intervals post infection. At day 7 post reinfection, blood and organs were analyzed. (A) Comparison between infected (right panels) and age-matched control mice (left panels), of the CDR3 TRD repertoire of blood samples at day 0, 21, 80 post-infection (d0, d21, d80) and at day 7 post-reinfection (d7R), and from organs at d7R. (Upper panels) TRDV usage distribution, (Middle panels) CDR3 length in nucleotides (including the codons for C-start and F-end residues), each dot represents the weighted mean of an individual sample. (Lower panels). (Lower panels) Number of N additions, each dot represents the weighted mean of an individual sample. (B) Percentage of unique clonotypes required to account for 50% of the total repertoire in infected or age-matched control mice. Statistical test was 1-way ANOVA.
(TIF)

**S6 Fig. Expansion of private CDR3δ repertoire following MCMV infection.** TCRα-/- mice (n = 5) were infected with MCMV. Age-matched uninfected TCRα-/- mice (n = 3) injected with medium were used as controls. Blood was drawn at different time intervals post-infection and at d7 post-reinfection. Clonotype tracking stackplots are shown with detailed profiles for top 100 clonotypes, as well as collapsed ("Not-shown" in light gray) and non-overlapping (dark gray) clonotypes found in blood of MCMV-infected mice (A) or age-matched control mice (B). Clonotypes are colored by the peak position of their abundance profile.
(TIF)

**S7 Fig. Tracking of shared clonotypes after reinfection in blood and organs.** TCRα-/- mice (n = 5) were infected with MCMV and reinfected 3 months later. At day 7 post reinfection (d7R), blood and organs were analyzed. (Upper panels) Clonotype tracking stackplots for the infected mice: detailed profiles for top 100 clonotypes, as well as collapsed ("NotShown" in light gray) and non-overlapping (dark gray) clonotypes found in blood and organs at d7R. Clonotypes are colored by the peak position of their abundance profile. The colors are matched for each mouse samples but not between the different mice. (Lower panels) Overlap frequencies of the D7R repertoire compared to the indicated column. Each dot corresponds to one pair comparison and one mouse. Statistical test was 1-way ANOVA.
(TIF)

**S8 Fig. Comparison of T cell subtypes in γδ from the spleen of long-term infected and age-matched uninfected TCRα-/- mice.** Percentages of γδ naïve (CD44-CD62L+), TCM (CD44+CD62L+) and TEM (CD44+CD62L-) in the spleen of 6 infected and 7 age-matched uninfected mice are shown, as well as the mean+/- SEM. One way ANOVA for comparison.
(TIF)

**S9 Fig. Repartition of γδ T cell memory subtypes in γδ-bearing CD3ε-/- mice infected with MCMV and in d92-infected TCRα-/- mice.** (Upper panels) Percentages of γδ TCM and TEM were determined in organs from 5 d92-MCMV infected TCRα-/- mice. (Lower panels) Long-term MCMV-induced γδ T cells were sorted from the spleen of TCRα-/- mice and transferred into CD3ε-/- mice that were subsequently infected with MCMV. Percentages of γδ TCM and TEM were determined in organs from 8, γδ bearing CD3ε-/- mice that had survived until d130. The proportion of indicated subtypes within total TCM and TEM is shown by shades of grey.
(TIF)

**S10 Fig. γδ TCM display higher survival and proliferative potential than γδ TEM.** (A, top) Experimental scheme. Total γδ T cells were sorted from the spleen of 3 months MCMV-infected TCRα-/- mice. γδ TCM and TEM subsets were sorted and transferred in CD3ε-/- mice (200000 cells/host). (A, lower left) Data represent percentages of γδ TCM and TEM among blood lymphocytes in individual mice, 3 weeks after transfer. (A, lower right) 3 weeks after transfer, mice were infected with MCMV concomitantly to control CD3ε-/- mice ($2.10^3$ PFU). Two experiments were performed with concordant results and were pooled in the figure. Horizontal bars show the means +/- SEM (Mann Witney) (B) Splenocytes from 3 months infected mice were labelled with CTY and cultured with or without IL-15 (200 ng/μl) for 3 days. (B, right) Percentages of viable cells among γδ T cells cultured in the absence or presence of IL-15. (B, middle) Percentages of proliferative γδ TCM or TEM after 3 days of culture with IL-15 (B, right) Representative histograms of cellular proliferation for TCM (light grey) and TEM (dark grey) from d92-infected TCRα-/- mice. Data are representative of 3 independent experiments. Images of mice and syringes were drawn by using pictures from Servier Medical Art. Servier Medical Art by Servier is licensed under a Creative Commons Attribution 3.0 Unported License (https://creativecommons.org/licenses/by/3.0/).
(TIF)

**S11 Fig. Single cell profiling of splenic γδ T cells from uninfected and MCMV-infected mice.** (A) UMAP projection of sample-originated cells. Cells from each sample were highlighted in red on the same UMAP embedding. UMAP algorithm was performed on the combined LSI scATAC-seq and scRNA-seq as described in Mat & Meth section. (B) Heatmap illustrating the top 10 up-regulated differentially expressed genes within clusters C1 to C8. Values are represented using the scaled of the mean log-normalized gene expression across cells among clusters (column). Row Z-scores are constrained to an upper bound of 2 and a lower bound of -2. (C) The heatmap shows the scaled of the mean log-normalized gene score matrix (prediction of a gene expression based on the accessibility) across cells among clusters (column) for a list of selected genes (row), categorized by 'functionality'. Column clustering was performed using Euclidean distance and the Ward.D2 method. Row Z-scores are constrained to an upper bound of 2 and a lower bound of -2. (D) Volcano plot representing the top 10 of up- and down-regulated genes (labeled red dots) between C5 and C8, based on the gene expression matrix. The dashed red line indicates the FDR $\leq$ 0.05. (E) Volcano plot representing the top 10 of up- and down-regulated genes (labeled red dots) between C5 and C6, based on the gene expression matrix. The dashed red line indicates the FDR $\leq$ 0.05.
(TIF)

**S1 Data. (Tab A-H). Raw data used to build Figures.**
(XLSX)

**S2 Data. (Tab A-H). Raw data used to build Supplementary Figures.**
(XLSX)

**S1 Table. CDR3aa Levenshtein distance summary.** Table summarizing the clustering of CDR3aa across the Levenshtein distance. This table describes for each CDR3aa, the mouse identifier, the cluster identifier, the number of connections (degree) and the total size of the cluster as well as information about the groups.
(XLSX)

**S2 Table. Top 10 gene expressions.** List of the 10 genes significantly up regulated by cluster, compared with the others for gene expression (GEX) and chromatin accessibility (ATAC).
(XLSX)

**S3 Table. Gene expression differential analysis.** Differential gene expression statistics by cluster (GEX)
(XLSX)

**S4 Table. Gene score differential analysis.** Differential gene score statistics by cluster (ATAC)
(XLSX)

**S5 Table. Gene expression differential analysis between C5 and C8.** Differential gene expression statistics comparing Cluster 5 and Cluster 8 (GEX)
(XLSX)

**S6 Table. Gene expression differential analysis between C5 and C6.** Differential gene expression statistics comparing Cluster 5 and Cluster 6 (GEX)
(XLSX)

**S7 Table. Gene Pathway differential analysis.** Functional analyses of differential gene lists (up and down separated) on Reactome database.
(XLSX)

## Acknowledgments

We thank Isabelle Douchet, Andrea Boizard-Moracchini and Coline Gentil at Immunoconcept for technical assistance during mice dissection and preparation of lymphocytes. We are grateful for the contribution of Parean Biotechnologies in single cell data analyses, whose members are engaged in the "HORUS" program (Casting Light on HOst-cytomegaloviRUs interaction in Solid organ transplantation) funded by HORIZON Europe. We thank the CNRS UAR3427, INSERM US05, TBM Core, FACSility and ONE CELL plateforms.

## Author Contributions

**Conceptualization:** David Vermijlen, Julie Déchanet-Merville, Myriam Capone.

**Data curation:** Nathalie Yared, Maria Papadopoulou, Pierre Barennes, Hang-Phuong Pham, Valentin Quiniou, Atika Zouine, Xavier Gauthereau.

**Formal analysis:** Nathalie Yared, Maria Papadopoulou, Pierre Barennes, Hang-Phuong Pham, Valentin Quiniou, Sonia Netzer, Laure Burguet.

**Funding acquisition:** Julie Déchanet-Merville, Myriam Capone.

**Investigation:** Maria Papadopoulou, Sonia Netzer, Hanna Kaminski, Laure Burguet, Lea Mora-Charrot, Benoit Rousseau, Julien Izotte, Myriam Capone.

**Methodology:** Nathalie Yared, Maria Papadopoulou, Sonia Netzer, Amandine Demeste, Pacôme Colas, Xavier Gauthereau, Myriam Capone.

**Project administration:** David Vermijlen, Julie Déchanet-Merville, Myriam Capone.

**Resources:** Pierre Barennes, Hang-Phuong Pham, Valentin Quiniou, Julie Déchanet-Merville, Myriam Capone.

**Supervision:** David Vermijlen, Julie Déchanet-Merville, Myriam Capone.

**Validation:** Nathalie Yared, Maria Papadopoulou, Sonia Netzer, David Vermijlen, Julie Déchanet-Merville, Myriam Capone.

**Visualization:** Nathalie Yared, Maria Papadopoulou, Pierre Barennes, Valentin Quiniou, Sonia Netzer, Atika Zouine, Julie Déchanet-Merville, Myriam Capone.

**Writing – original draft:** Nathalie Yared, Maria Papadopoulou, David Vermijlen, Julie Déchanet-Merville, Myriam Capone.

**Writing – review & editing:** Pierre Barennes, Valentin Quiniou.

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
