## [Decision Letter · Decision Letter 0]

12 Sep 2022

Dear Dr CAPONE,

Thank you very much for submitting your manuscript "Long-lived central memory γδ T cells confer protection against murine cytomegalovirus reinfection" for consideration at PLOS Pathogens. As with all papers reviewed by the journal, your manuscript was reviewed by members of the editorial board and by several independent reviewers. In light of the reviews (below this email), we would like to invite the resubmission of a significantly-revised version that takes into account the reviewers' comments.

Although reviewers #1 and #2 classified their responses as "minor", the concerns of Reviewer #1 in particular are significant and align with the major issues raised by reviewer #3 and with some of my own thoughts. While all of the comments must be considered, the authors should pay particular attention to concerns by all 3 reviewers about the conclusion that gd T cells are true “memory” cells at the late time points. Reviewers were concerned about the numbers of these cells and the modest degree of recall expansion vs the possibility that results are based on altered function. Moreover, there is an apparent lack of clonal expansions after re-challenge and it is not clear what this means with regards to the classical definition of memory T cells. I agree with the reviewers on these points. These concepts represent the core conclusions and novelty of this work and the authors should clearly demonstrate whether T cells from previously infected mice are memory T cells in the classical sense or something else. This is difficult in the intact animal since the virus is presumably persistently replicating in this model and new gd T cells can be made throughout the experiment, which could contribute to the private clones. The adoptive transfer setting may offer the ideal tool for examining some of these issues (i.e. what happens if they are allowed to rest? do the TCMs retain better proliferative function? Do TEMs have better immediate effector function?, is there clonal expansion on re-challenge?). In addition, the authors describe the gd T cells as having expanded after infection or rechallenge, but the data appear to show that cells are lost from the blood and then recover over time. The numbers in Figure 1 really only seem to increase from baseline in the lungs, but this could represent a redistribution and not an expansion. Additionally, reviewers #2 and #3 had concerns about the interpretation of the antibody blocking experiments shown in Figure 6 and Reviewer #3 points out an important missing control for this experiment. Likewise, reviewer #3 raises an important question about NK cells in these experiments, which should be discussed since they will be activated by m157 in this model.

We cannot make any decision about publication until we have seen the revised manuscript and your response to the reviewers' comments. Your revised manuscript is also likely to be sent to reviewers for further evaluation.

Sincerely,

Christopher M. Snyder, Ph.D.

Guest Editor

PLOS Pathogens

Klaus Früh

Section Editor

PLOS Pathogens

Kasturi Haldar

Editor-in-Chief

PLOS Pathogens

orcid.org/0000-0001-5065-158X

Michael Malim

Editor-in-Chief

PLOS Pathogens

orcid.org/0000-0002-7699-2064

Although reviewers #1 and #2 classified their responses as "minor", the concerns of Reviewer #1 in particular are significant and align with the major issues raised by reviewer #3 and with some of my own thoughts. While all of the comments must be considered, the authors should pay particular attention to concerns by all 3 reviewers about the conclusion that gd T cells are true “memory” cells at the late time points. Reviewers were concerned about the numbers of these cells and the modest degree of recall expansion vs the possibility that results are based on altered function. Moreover, there is an apparent lack of clonal expansions after re-challenge and it is not clear what this means with regards to the classical definition of memory T cells. I agree with the reviewers on these points. These concepts represent the core conclusions and novelty of this work and the authors should clearly demonstrate whether T cells from previously infected mice are memory T cells in the classical sense or something else. This is difficult in the intact animal since the virus is presumably persistently replicating in this model and new gd T cells can be made throughout the experiment, which could contribute to the private clones. The adoptive transfer setting may offer the ideal tool for examining some of these issues (i.e. what happens if they are allowed to rest? do the TCMs retain better proliferative function? Do TEMs have better immediate effector function?, is there clonal expansion on re-challenge?). In addition, the authors describe the gd T cells as having expanded after infection or rechallenge, but the data appear to show that cells are lost from the blood and then recover over time. The numbers in Figure 1 really only seem to increase from baseline in the lungs, but this could represent a redistribution and not an expansion. Additionally, reviewers #2 and #3 had concerns about the interpretation of the antibody blocking experiments shown in Figure 6 and Reviewer #3 points out an important missing control for this experiment. Likewise, reviewer #3 raises an important question about NK cells in these experiments, which should be discussed since they will be activated by m157 in this model.

Reviewer's Responses to Questions

**Part I - Summary**

Reviewer #1: An interesting and timely study. However, this reviewer is not convinced by some of the conclusions derived from this data, most notably the notion that the observed effects constitute proof of a memory response when functional changes within the gd population and TCR repertoire changes between primary and secondary response are relatively modest.

Reviewer #2: Yared et al. address the “memory like nature” of γδ T cells in a model of murine cytomegalovirus infection. The paper is a follow up of a study from the same group demonstrating that γδ T cells, in the absence of classical adaptive immune cells (αβ T cells and B-cells), are sufficient to confer protection of mice from an otherwise lethal infection (Kharailla et al, PLoS Pathog. 2015 Mar; 11(3): e1004702). The new paper addresses the possibility of γδ T cell memory formation in MCMV infection. MCMV infection serves as a model of HCMV infection for which a role of γδ T cells in infection control was originally described by the same group. In general, the experiments are carefully designed and described but there are some questions which should be addressed before publication. In addition, there are some minor errors and interpretations of results which in my eyes are not fully supported by the data and may be discussed.

The paper is of considerable interest since i) MCMV is a well-established animal model in which virus control by αβ T-, B- and NK cells has been studied in quite some detail including analysis of immunological memory but not the possibility of γδ T cell memory. ii) γδ T cells have raised considerable interest as immunotherapeutic tool including the potential use of γδ T cells to control (HCMV) infection.

Reviewer #3: Capone et al suggest that memory gd T cells can protect mice against MCMV, and this protection is found in the central memory compartment. Overall, the study is quite descriptive. The roughly 2-3 fold expansion in gd T cell number during primary or secondary expansion is relatively modest. The key experiments are found in Fig 8 and 9, which demonstrate in an adoptive transfer setting the protective ability of Tcm but not Tem cell in the gd population. However, no mechanism behind this observation is shown. Are the Tcm producing more IFNg or become more cytolytic during secondary exposure. Furthermore, if the authors could demonstrate how gd T cells are seeing MCMV components (what is the ligand?), this would add significant novelty to the study. Otherwise, the overall analysis of bulk gd T cell populations dampens my enthusiasm for this manuscript.

**Part II – Major Issues: Key Experiments Required for Acceptance**

Reviewer #1: (No Response)

Reviewer #2: (No Response)

Reviewer #3: Are the authors concerned that MCMV goes latent? Is there reactivation during the 3 months between virus injections?

In Fig 1-2, are the cells measured during the secondary infection the same cells that have seen virus the first time? Or new cells responding? Or cells that have been primed during latency?

How is it that the authors are able to challenge the same mice with MCMV twice? Are there no neutralizing antibodies against MCMV?

Regarding the enhanced clearance of the virus observed in Fig 1B, can the authors attribute this to gd T cells? Or is it simply neutralizing Abs, or even memory NK cells (which have previously been proposed in the setting of CMV infection)?

The antibody blocking experiments should probably go into supplement, as I’m not sure how to interpret the findings. Plus, a key control that is missing is gd TCR blockade in naïve mice infected with MCMV. Would the authors see equally high titers in this setting as well?

The authors should determine the antigen-specificity of the gd T cells. This may help explain why the gd T cells contract and not inflate like some T cell populations during MCMV.

If there is clonal expansion, I’m confused why there is no enrichment of specific clones following secondary infection (in fig 7B).

**Part III – Minor Issues: Editorial and Data Presentation Modifications**

Reviewer #1: - The statistical analysis in Fig.1B is not clear. What kind of post-hoc test was used for multiple comparisons, and what reference was used? In Fig.1C, day 3 was used as reference - why? There is no statistical analysis of the MCMV copy data

Fig.1B and Fig.2: the authors should comment on why there is such a sharp peak on day 1 in the secondary response in the liver that then declines by days 3-7. Any theory?

- Data presentation in general: the authors may want to consider showing an overlay of primary and secondary responses as line or bar diagram to make comparisons between different responses easier, rather than showing a continuous line throughot primary and secondary responses.

- Fig.S1: there are only relatively minor changes in the constitution of the gd compartment during the secondary infection. Doesn't this lack of clear functional changes within the gd population reflect a lack of a bona fide memory response, other than an expansion?

- Fig.5B: it's surprising to see the authors completely dismiss the effect of FTY720 treatment in the liver when the p value is at 0.055. It could be argued that the experiment was simply underpowered and that its design prevented this p value from falling below 0.05. A statement like "no statistical differences were evidenced" (lines 229-230) is simply not good enough to discuss the potential biological effect of FTY720 treatment.

- Fig.7A: this figure should show data from all mice not just a selection of some.

Reviewer #2: The paper is of considerable interest since i) MCMV is a well-established animal model in which virus control by αβ T-, B- and NK cells has been studied in quite some detail including analysis of immunological memory but not the possibility of γδ T cell memory. ii) γδ T cells have raised considerable interest as immunotherapeutic tool including the potential use of γδ T cells to control (HCMV) infection.

Figure 1. For some individuals no virus could be detected (I guess 1 means between 0 and 1). Did the authors find any correlation between negativity/low virus load and any of the other parameters? The authors should address and discuss this, even if it might complicate the statistical analysis (also for the following figures).

Figure 5. The data are clear and underline the role of tissue-resident cells. Did the authors or others also perform similar experiments for αβ T cells or wild type mice? If yes, it would be interesting to learn about it in the manuscript.

Figure 6. The authors treat animals with the γδ TCR-modulating antibody GL3 and find a clear reduction in virus control in treated mice. They interpret this as evidence for a role of the TCR in virus control. Although the data are suggestive, this interpretation goes too far. Without providing further evidence, one must assume that the antibody merely binds to γδ T cells and that a general impairment of γδ T cell function will be caused by antibody-induced TCR modulation. Therefore, this experiment provides no evidence for a direct role of the γδ TCR in virus control. It is just a very nice confirmation of the results obtained with the deficient mice and the adoptive transfer experiments.

Figure 7. The authors should explain the significance of an overlapping TCR-repertoire. Actually, I find the entire part difficult to understand. I also found the remark quite interesting that injection of medium caused also some effects. The authors may consider and discuss that changes in the repertoire reflect a type of stress response of the “adaptive/adaptate” γσ T cells and not a specific response to MCMV. In such a setting the TCRs would serve as a kind of bar code of expanded clones rather than as indicators of an antigen-specific clonal expan

---

## [Decision Letter · Decision Letter 1]

28 Mar 2024

Dear Dr CAPONE,

Thank you very much for submitting your manuscript "Long-lived central memory γδ T cells confer protection against murine cytomegalovirus reinfection" for consideration at PLOS Pathogens. As with all papers reviewed by the journal, your manuscript was reviewed by members of the editorial board and by several independent reviewers. In light of the reviews (below this email), we would like to invite the resubmission of a significantly-revised version that takes into account the reviewers' comments.

Given the time since the initial submission, we had to recruit 2 new reviewers. These reviewers have both attempted to evaluate the manuscript fairly and did identify some areas that should be addressed. However, I think that most of their major concerns can be addressed by revisions to the text or perhaps the order of the figures. Overall, addressing the identified major issues won't substantially alter the core conclusions of the study. However, the caveats raised may be valid and therefore the authors should focus on revising the manuscript to be very clear about the limits and caveats of the experiments and data, to be more clear about the central novel interpretations and the key data that supports these conclusions, and perhaps consider changing the order in which information is presented. Additionally, reviewers 4 and 5 both identified concerns with Figure 3B and Figure 4B, which are presented to suggest that gd T cells may impact viral loads. Although the authors may wish to address these concerns, I’m not sure that these data are absolutely required (at least within the main figures) for the major conclusions of the manuscript. The adoptive transfer experiments in Figures 7 and 8, as well as the GL3 treatments in Fig 6, seem to be the most conclusive experiments showing that gd T cells reduce viral loads. At the least, I agree with both reviewers that these viral load data aren't easy to interpret right now.

We cannot make any decision about publication until we have seen the revised manuscript and your response to the reviewers' comments. Your revised manuscript is also likely to be sent to reviewers for further evaluation.

Sincerely,

Christopher M. Snyder, Ph.D.

Guest Editor

PLOS Pathogens

Klaus Früh

Section Editor

PLOS Pathogens

Michael Malim

Editor-in-Chief

PLOS Pathogens

orcid.org/0000-0002-7699-2064

Given the time since the initial submission, we had to recruit 2 new reviewers. These reviewers have both attempted to evaluate the manuscript fairly and did identify some areas that should be addressed. However, I think that most of their major concerns can be addressed by revisions to the text or perhaps the order of the figures. Overall, addressing the identified major issues won't substantially alter the core conclusions of the study. However, the caveats raised may be valid and therefore the authors should focus on revising the manuscript to be very clear about the limits and caveats of the experiments and data, to be more clear about the central novel interpretations and the key data that supports these conclusions, and perhaps consider changing the order in which information is presented. Additionally, reviewers 4 and 5 both identified concerns with Figure 3B and Figure 4B, which are presented to suggest that gd T cells may impact viral loads. Although the authors may wish to address these concerns, I’m not sure that these data are absolutely required (at least within the main figures) for the major conclusions of the manuscript. The adoptive transfer experiments in Figures 7 and 8, as well as the GL3 treatments in Fig 6, seem to be the most conclusive experiments showing that gd T cells reduce viral loads. At the least, I agree with both reviewers that these viral load data aren't easy to interpret right now.

Reviewer's Responses to Questions

**Part I - Summary**

Reviewer #2: In general, I am fine with the answers to my previous comments and the revision. Just two points.

Fig S2B. Please enlarge the symbols or give them different colors.

Fig. 5 The authors may discuss.

Fig. 5 B and D. Despite that gd T cells of the antibody treated animals can be activated in a TCR-independent manner (PMA/Iono), TCR-antibody might affect effector function/survival of the primed/trained cells triggered by innate/cytokine receptors.

Fig. 5D gdT cells may be primed/trained in a TCR independent manner (first two weeks) and but primed/trained cells require TCR-mediated signals to execute effector functions in the third week (Fig. 5D). This would be in line with a MCMV TCR-specific response. Alternatively, the TCR antibody could provide a TCR-mediated signal interfering with effector functions triggered by innate/cytokine receptors (see above).

Reviewer #4: The manuscript by Yared et al. describes an interesting adaptive-like feature of gd T cells in response to MCMV infection, extending on two previous publications appearing almost at the same time. In their manuscript, the authors elaborate on the crucial question whether immunological memory is generated in the gd T cell compartment. One of the previous publications already suggested a major difference in protecting immunodeficient mice when gd T cells from infected mice were used in transfers. In the current manuscript, the authors then start to characterize the features and functions of the memory-like gd T cells.

The major problem in the characterization is the assumption that well-established markers for memory subpopulations of CD8+ aß T cells, particularly TEM and TCM are transferable to the gd lineage. Without comparing the respective populations side by side, this is a dangerous misconception, in my opinion, as it paves the way for future simplifications in following studies.

The authors already show in the experiment for Figure 1, for example, that gd cells do not compare to the classical categories, also not regarding expansion and contraction, at least to a much lesser extent. ”TEM” cells would express CX3CR1 and KLRG1, in naïve mice most of them don´t, however. A detailed comparison by the transcriptional landscapes would be helpful to dissect the gd subpopulation and compare them to the CD8 aß subpopulation. The analysis in sFig.2 goes in this direction. As this analysis does not differentiate the subpopulations, it is not able to clarify this critical issue, however.

Strangely, a significant attempt to differentiate the subpopulations is made in Figure 9, most likely because of the previous review process. These “new” data should be shown and discussed after Figure 1 and include a much more detailed bioinformatic analysis, possibly including a comparison with CD8 aß TEM and TCM as they can easily be obtained from deposited data sets.

Reviewer #5: I has a chance to review the article “Long-lived central memory γδ T cells confer protection against murine cytomegalovirus reinfection” by the group of M. Capone that was resubmitted to PLOS Pathogens upon in initial review.

Since one of the primary reviewers appears to be unavailable, I was asked to step in during the secondary review. In fairness to the authors, my review does not address questions that were not raised by reviewers during primary review, but in fairness to the process, I had to go through a detailed assessment of the article and raise objections where I identified concerns about data and their interpretation. Overall, I think that the study is interesting and relevant to the readers of PLOS Path., but that several item need to be addressed as major concerns that require additional experimental evidence.

**Part II – Major Issues: Key Experiments Required for Acceptance**

Reviewer #2: see part I

Reviewer #4: The analysis in Fig. 2 is important and the data appear clear in showing that most tissue-based gd T cells 3m after infection appear in the blood. It would be important to strengthen this finding by a control experiment for tissue-resident gd T cells in the same experimental setup, even from the same mice. Dermal or intestinal gd T cells would be very good controls.

In Fig.3 the strong reduction of TCM in lung, liver and blood after FTY720 treatment is interpreted as block of lymph node egress. This statement is not valid unless data are shown that TCM are, in fact, residing and, after treatment, accumulating in lymph nodes. They might be otherwise affected. The data in Fig. 3b should not be shown here as there is no clear conclusion possible.

Figure 4 shows a somewhat faster and to some extend stronger response in secondary infections. The extent is, however, much less pronounced when compared to CD8 ab responses. This should be mentioned somewhere in the manuscript.

The changes in viral load, i.e. the potentially better protective function in the “memory” situation are not very convincing and the effect of expanding NK cells as well as the potential role of antibodies preclude a solid conclusion at the current state of the manuscript.

In Figure 5 the effects of anti-GL3 treatment are clear regarding control of viral load. The interpretation that specific recognition by the gdTCR cannot be drawn by this experiment. The persisting gd T cells after GL3 treatment might just be severely impaired in their function. The control experiment showing IFNg production after 3h ex vivo stimulation is not sufficient. How about other effector molecules like granzymes and perforin?

The positive effects of transfer of gd T cells from long-term infected mice is extending previous findings described in PMID: 25658831 This should at least be mentioned and the additional gain of insight should be worked out.

Figure 8 and 9 are the most crucial parts in the manuscript as a clear protective role of TCM-like cells is shown by adoptive transfer and analyzed in detail by the scRNA experiment. The weakness is, however, that here only splenic gd T cell populations are used for analysis as well as transfer. Given the potential importance of virus persistence / latency in lung and liver, only “TEM-like cells” from the respective organs might confer protection upon transfer. This would lead to a fundamentally different conclusion, in my opinion.

Figure 9 needs more attention in the text, and the analysis appears rather rudimentary. At least a description of the 8 UMAP clusters should be presented, for example by displaying the top up- or downregulated genes and signature genes as bubble or violin plots

Reviewer #5: The list below is in order of appearance in the text:

1. In Fig. 3B, the authors were requested address the weak stat power of their experiment, resulting in a borderline p value. They addressed this request by altering the interpretation of results, which is not an appropriate course of action: they should instead repeat the assay, generate additional evidence and define if the FTY720 effects on viral loads are significant and reproducible or not.

2. In Figure 4B, the authors show viral loads as fold changes relative to Day(0) upon primary or secondary infection. They start from a value of 1 in either scenario to show a drastic increase in primary, but not in secondary infection. The problem is that at day 0, the mice were MCMV naïve and the value had to be 0. Therefore, any increase would end up as an infinite-fold increase. The authors have obviously used an arbitrary value for the day 0 of the primary infection, which is not immediately obvious. This needs to be explained more accurately.

3. In reference to the first comment by Rev#3, the authors have indeed produced no data that the MCMV indeed establishes latency, rather than persisting in their system. Since it is well known that CD4 T cells are required for MCMV control in salivary glands, it is reasonable to assume that their model may be affected by MCMV persistence in this organ (and spurious dissemination of virus to other organs from this focus of persistence). At a minimum, the authors need to perform plaque assays on SG homogenates at a late time post infection. Ideally, they should perform it on all organs. The conclusions of this assay will not preclude publication, but they will substantially influence data interpretation (e.g. they may indicate that there is no additional spike in viral loads upon reinfection, because the virus is lytically persistent even before reinfection). Hence, this evidence is critically important.

**Part III – Minor Issues: Editorial and Data Presentation Modifications**

Reviewer #2: see part I

Reviewer #4: The y-axis in Fig. 5 c and d should be labeled MCMV copies / ug DNA

Reviewer #5: 1. In Figure 5D, the authors now show that the depletion of gd T cells prior to MCMV primary infection results in no effects on viral titer at dpi 7. Nevertheless, the effects are prominent if the depletion occurs later with a similar waiting time of 7 days. How can this be? The authors need to discuss these data. Also, the marking next to the y axis in Fig. 5D spells ADN, instead of DNA.

2. The methods used to perform the cluster analysis in Fig. 6C are rather confusing. More detail and careful wording may be sufficient to address this issue.

3. In Fig. S11B, it is not clear what is shown by bubble size. Are these cells from infected vs. uninfected mice? Did they compare the nr. of genes per cluster, or the aggregate gene expression levels of genes belonging to a pathway/cluster? Or something else?

PLOS authors have the option to publish the peer review history of their article (<a href="https://journals.plos.org/plospathogens/s/ed

---

## [Editor Report · Decision Letter 2]

12 Jun 2024

Dear Dr CAPONE,

We are pleased to inform you that your manuscript 'Long-lived central memory γδ T cells confer protection against murine cytomegalovirus reinfection' has been provisionally accepted for publication in PLOS Pathogens.

Best regards,

Christopher M. Snyder, Ph.D.

Guest Editor

PLOS Pathogens

Klaus Früh

%CORR_ED_EDITOR_ROLE%

PLOS Pathogens

Michael Malim

Editor-in-Chief

PLOS Pathogens

orcid.org/0000-0002-7699-2064

Thank you for working through the review process. We feel that this represents a significant advance to the field. There are small formatting errors noted below that should be corrected in the final article.

Minor edits for final manuscript:

Line 296 – appears to have a citation formatting error (Sell, 2015 #44)

Suppl Fig 10B (leftmost panel) appears to have a formatting issue with the labels on the X-axis. The legend and text suggest that there should only be 2 groups but there are 3 symbols included on the x-axis. In addition, there is a typo in the legend for this figure. Line 1194 refers to the leftmost panel, but the text states that it is referring to the right panel.
---

## [Editor Report · Acceptance letter]

2 Jul 2024

Dear Dr CAPONE,

We are delighted to inform you that your manuscript, "Long-lived central memory γδ T cells confer protection against murine cytomegalovirus reinfection," has been formally accepted for publication in PLOS Pathogens.

Best regards,

Michael Malim

Editor-in-Chief

PLOS Pathogens

orcid.org/0000-0002-7699-2064